# Centroid Approximation for Bootstrap

## Abstract

Bootstrap is a principled and powerful frequentist statistical tool for uncertainty quantification. Unfortunately, standard bootstrap methods are computationally intensive due to the need of drawing a large i.i.d. bootstrap sample to approximate the ideal bootstrap distribution; this largely hinders their application in large-scale machine learning, especially deep learning problems. In this work, we propose an efficient method to explicitly *optimize* a small set of high quality "centroid" points to better approximate the ideal bootstrap distribution. We achieve this by minimizing a simple objective function that is asymptotically equivalent to the Wasserstein distance to the ideal bootstrap distribution. This allows us to provide an accurate estimation of uncertainty with a small number of bootstrap centroids, outperforming the naive i.i.d. sampling approach. Empirically, we show that our method can boost the performance of bootstrap in a variety of applications.

## 1 Introduction

Bootstrap is a simple and principled frequentist uncertainty quantification tool and can be flexibly applied to obtain data uncertainty estimation with strong theoretical guarantees (Hall et al., 1988; Austern & Syrgkanis, 2020; Chatterjee et al., 2005; Cheng et al., 2010). In particular, when combined with the maximum likelihood estimator or more general M-estimators, bootstrap provides a general-purpose, plug-and-play non-parametric inference framework for general probabilistic models without case-by-case derivations; this makes it a promising frequentist alternative to Bayesian inference.

However, the standard bootstrap inference is highly expensive in both computation and memory as it typically requires drawing a large number[1] of i.i.d. bootstrap particles (samples) to obtain an accurate uncertainty estimation. In the context of this paper, as each bootstrap particle/sample/centroid is a machine learning model, we might directly call a model as particle/sample/centroid. With a small number of particles, bootstrap may perform poorly. As a consequence, when applied to deep learning, we need to store a large number of neural networks and feed the input into a tremendous number of networks every time we make inference, which can be quite expensive and even unaffordable for deep learning problems with huge models[2]. For example, in autonomous driving applications, our device can only store a limited number of models and we need to make decisions within a short time, which makes the standard bootstrap with a large number of models no more feasible. Typical ensemble methods in deep learning, such as Lakshminarayanan et al. (2017); Huang et al. (2017); Vyas et al. (2018); Maddox et al. (2019); Liu & Wang (2016), can only afford to use a small number (e.g., less than 20) of models.

Therefore, to make bootstrap more accessible in modern machine learning, it is essential to develop new approaches that break the key computation and memory barriers mentioned above. This paper aims to improve the bootstrap when the resource at *inference* is limited. We are motivated to consider the following problem:

*How to improve the accuracy of bootstrap inference when the number of particles is limited?*

We attack this challenge by presenting an efficient centroid approximation for bootstrap. Our method replaces the i.i.d. bootstrap particles with a set of carefully optimized *centroid particles* that are

---

[1]For example, thousands of, as suggested by Statistics textbooks such as Wasserman (2013).

[2]While training cost is an extra burden, it is small compared with the cost of making prediction as we only need to train the model once but make countless predictions at deployment.

guaranteed to provide an accurate and compact approximation to the ideal bootstrap distribution so that only a smaller number of particles is needed to obtain good performance.

Our method is based on minimizing a specially designed objective function that is asymptotically equivalent to the Wasserstein distance between the ideal bootstrap distribution and the particle distribution formed by the learned centroids. During the training, each centroid adjusts its location being aware of the locations of the others so that centroids are diversified and well distributed on the domain. Our method is similar to doing K-means on the ideal bootstrap distribution, finding K representative centroids that well represent K separate parts of the target distribution's domain in an optimal way. As centroids are optimized to better approximate the distribution, our approach naturally improves over the vanilla bootstrap with i.i.d. particles. See Figure 1 for illustration.

Empirically, we apply the centroid approximation method to various applications, including confidence interval estimation (DiCiccio et al., 1996), bootstrap method for contextual bandit (Riquelme et al., 2018), bootstrap deep Q-network (Osband et al., 2016) and bagging method (Breiman, 1996) for neural networks. We find that our method consistently improves over the standard bootstrap.

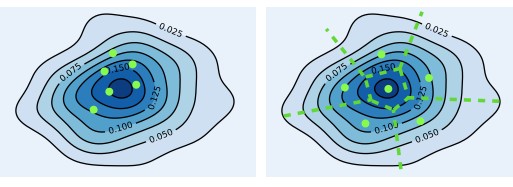

Figure 1: The solid lines represent the density of the target distribution. Left figure: Typical i.i.d. particles that are randomly distributed on the domain. Right figure: The learned diversified centroids that are well distributed on the domain. The centroids partition the domain into several disjoint regions (separated by the dashed lines in the figure) and each centroid can be viewed as the 'K-means' center of the region it belongs to.

**Notation** We use $\|\cdot\|$ to represent the $\ell_2$ norm for a vector and the operator norm for a matrix. We denote the integer set $\{1, 2, ...., N\}$ by $[N]$. Given any $m$, we define the probability simplex $\mathcal{C}^m := \{[v_1, ..., v_m] \in \mathbb{R}^m : v_i \geq 0, \forall i \in [m] \text{ and } \sum_{i \in [m]} v_i = 1\}$. For a symmetric matrix $M$, we denote its minimal eigenvalue by $\lambda_{\min}(M)$. For a positive-definite matrix $M$, if $M = A^\top A$, then we denote $A$ by $M^{1/2}$. We denote the Wasserstein distance between two distribution $\rho_1$ and $\rho_2$ by $\mathcal{W}_2[\rho_1, \rho_2]$. We use $O$ and $o$ to denote the conventional big-O and small-o notation and use $O_p$ to denote the stochastic boundedness. We use $\overset{d}{\to}$ to denote convergence in distribution.

## 2 BACKGROUND

Suppose we have a model $f_\theta$ parameterized by $\theta$ in a parameter space $\Theta \subseteq \mathbb{R}^d$. Let $\{x_i\}_{i=1}^n \subset \mathcal{X}$ be a training set with $n$ data points on $\mathcal{X}$. Assume $\ell(x, f_\theta)$ is the negative log-likelihood of data point $x$ with model $f_\theta$. A standard approach to estimate $\theta$ is maximum likelihood estimator (MLE), which minimizes the negative log-likelihood function (loss) over the training set

$$\hat{\theta} = \arg \min_{\theta \in \Theta} \mathcal{L}(\theta), \qquad \mathcal{L}(\theta) = \sum_{i=1}^n \ell(x_i, f_\theta)/n.$$

Here the MLE $\hat{\theta}$ provides a point estimation without any information on the data uncertainty. Bootstrap is a simple and effective frequentist method to quantify the uncertainty. The bootstrap loss is a randomly perturbed loss defined as

$$\mathcal{L}_{\boldsymbol{w}}(\theta) = \sum_{i=1}^n w_i \ell(x_i, f_\theta)/n,$$

where $\boldsymbol{w} = [w_1, ..., w_n]^\top$ is a set of random weights of data points drawn from some distribution $\pi$. A typical choice of $\pi$ is the multinomial distribution with uniform probability, which corresponds to resampling on the training set with replacement. Given $\boldsymbol{w}$, one can calculate its associated bootstrap particle by minimizing the bootstrap loss:

$$\hat{\theta}_{\boldsymbol{w}} = \arg \min_{\theta \in \Theta} \mathcal{L}_{\boldsymbol{w}}(\theta). \tag{1}$$

Let $\rho_\pi$ be the distribution of $\hat{\theta}_{\boldsymbol{w}}$ when $\boldsymbol{w} \sim \pi$. Bootstrap theory indicates that we can quantify the data uncertainty of $\theta$ or any function $g(\theta)$ using $\rho_\pi$. We call $\rho_\pi$ the ideal bootstrap distribution and it is the main object we want to approximate.

Denote $\delta_\theta$ as the delta measure centered at $\theta$. Standard bootstrap method approximates $\rho_\pi$ by the particle distribution $\hat{\rho}_\pi(\cdot) = \sum_{j=1}^m \delta_{\hat{\theta}_{w_j}}(\cdot)/m$ formed by $m$ i.i.d. particles $\{\hat{\theta}_{w_j}\}_{j=1}^m$, which can be obtained by drawing $m$ i.i.d. weights $\{w_j\}_{j=1}^m$ from $\pi$ and calculating each $\hat{\theta}_{w_j}$ based on (1). However, for deep learning applications, as discussed in the introduction, storing and making inference using a large number $m$ of bootstrap particles can be quite expensive. On the other hand, if $m$ is small, $\hat{\rho}_\pi$ tends to be a poor approximation of $\rho_\pi$. In this paper, we aim to improve the approximation of the particle distribution when $m$ is small.

## 3 METHOD

Our idea is simple. Instead of using i.i.d. particles, in which the location of each particle is independent from that of the others, we try to actively optimize the location of each particle so that particles are diversified, better distributed and eventually providing a particle distribution with improved approximation accuracy. A natural way to achieve this goal is to explicitly optimize a set of points $\{\theta_j\}_{j=1}^m$ (called centroids) jointly such that the Wasserstein distance between $\rho_\pi$ and the induced particle distribution is minimized:

$$\{\theta_j^*, v_j^*\}_{j=1}^m = \arg\min_{\theta_1,\ldots,\theta_m \in \Theta, \, [v_1,\ldots,v_m] \in \mathcal{C}^m} \mathcal{W}_2\left[\sum_{j=1}^m v_j \delta_{\theta_j}, \rho_\pi\right]. \tag{2}$$

Here we consider a Wasserstein distance $\mathcal{W}_2$ equipped with a special data-dependent distance metric $||\cdot||_D$ that will be introduced later in (5). We note that here we also optimize the probability weights $\{v_j\}_{j=1}^m$ of the centroids. Finding the optimal centroids and probability weights can be decomposed into two steps: the centroid learning phase and the probability weights learning phase, based on the facts in (3,4).

$$\mathcal{W}_2^2\left[\sum_{j\in[m]} v_j^* \delta_{\theta_j^*}, \rho_\pi\right] = J_\pi(\{\theta_j^*\}_{j=1}^m), \text{ where } J_\pi(\{\theta_j\}_{j=1}^m) := \mathbb{E}_{w\sim\pi}\left[\min_{j\in[m]}||\theta_j - \hat{\theta}_w||_D^2\right]. \tag{3}$$

Here (3) implies that, to find the optimal particle distribution in (2), we can start with the centroid learning phase where we only need to optimize the centroids. It can be achieved by minimizing $J_\pi(\{\theta_j\}_{j=1}^m)$, which is the averaged distance of bootstrap particles to their closest centroid. After we obtain the optimal centroids, the optimal probability weights can be learned by (4):

$$v_j^* = \tilde{v}_j^*/\sum_{s\in[m]}\tilde{v}_s^*, \text{ where } \tilde{v}_j^* = \mathbb{P}_{w\sim\pi}\left(j = \arg\min_{j\in[m]}||\theta_j^* - \hat{\theta}_w||_D^2\right). \tag{4}$$

Here $v_j^*$ is the proportion of bootstrap particles that are closest to the centroid $j$. We emphasize that the optimal solution to two-stage learning is guaranteed to be the global minimizer of the loss in (2) (see Lemma 3.1 and 3.2 in Canas & Rosasco (2012)).

However, the key issue is that the losses in both (2, 3) can not be computed in practice, as they require us to access $\rho_\pi$ (i.e., obtain $\hat{\theta}_w$ first in order to calculate the loss). To handle this issue, we seek an easy-to-compute surrogate loss. Our idea is based on the following observation. Assuming the size of training data is large, which is usually the case in deep learning, we can expect that $\theta_w$ will be centered around a small region[3]. It implies that we should search the centroid in this small region. Notice that when $\theta$ is close to $\hat{\theta}_w$, based on Taylor approximation, we have

$$\mathcal{L}_w(\theta) \approx \mathcal{L}_w(\hat{\theta}_w) + \nabla_\theta^\top \mathcal{L}_w(\hat{\theta}_w)(\theta - \hat{\theta}_w) + 1/2(\theta - \hat{\theta}_w)^\top \nabla_\theta^2 \mathcal{L}_w(\hat{\theta}_w)(\theta - \hat{\theta}_w)$$

$$\approx \mathcal{L}_\infty(\theta_0) + 1/2||\theta - \hat{\theta}_w||_D^2, \text{ where } ||V||_D^2 := V^\top \nabla_\theta^2 \mathcal{L}_\infty(\theta_0)V. \tag{5}$$

Here $\mathcal{L}_\infty(\theta) := \mathbb{E}_x \ell(x, f_\theta)$ denotes the population loss; $\theta_0$ is the minimizer of $\mathcal{L}_\infty(\theta)$. In (5), we use the facts[4] that $\nabla_\theta^\top \mathcal{L}_w(\hat{\theta}_w) = 0$; and with large training set, the empirical distribution $\sum_{i=1}^n \delta_{x_i}/n$ well approximates the whole data population, and hence the bootstrap resampling distribution, i.e., $\sum_{i=1}^n w_i \delta_{x_i}/n$ on the empirical distribution also well approximates the whole data population. This implies that $\mathcal{L}_w(\cdot) \approx \mathcal{L}_\infty(\cdot)$ and $\nabla_\theta^2 \mathcal{L}_w(\cdot) \approx \nabla_\theta^2 \mathcal{L}_\infty(\cdot)$. As the loss are close to each other, their minimizers are also close $\hat{\theta}_w \approx \theta_0$. Since $\mathcal{L}_\infty(\theta_0)$ is some (unknown) constant independent with $\theta$, we can replace the $||\theta_j - \hat{\theta}_w||_D^2$ in (3) by $\mathcal{L}_w(\theta_j)$ as it only adds some constant into the loss.

---

[3]This can be formally characterized by central limit theorem as discussed in Section 4.

[4]We defer the detailed analysis to Section 4.

Intuitively, we can expect that the centroid closest to $\hat{\theta}_{\boldsymbol{w}}$ is the one that gives the smallest loss on $\mathcal{L}_{\boldsymbol{w}}$. It motivates us to learn the centroids via the modified centroid learning phase:

$$\{\theta_j^*\}_{j=1}^m = \arg\min_{\theta_1,\ldots,\theta_m\in\Theta} \mathbb{E}_{\boldsymbol{w}\sim\pi}\left[\min_{j\in[m]}\mathcal{L}_{\boldsymbol{w}}(\theta_j)\right]. \tag{6}$$

Similarly, the optimal probability weights can be learned via the modified weight learning phase:

$$v_j^* = \tilde{v}_j^*/\sum_{s\in[m]}\tilde{v}_s^*, \text{ where } \tilde{v}_j^* = \mathbb{P}_{\boldsymbol{w}\sim\pi}\left(j\in\arg\min_{j\in[m]}\mathcal{L}_{\boldsymbol{w}}(\theta_j^*)\right). \tag{7}$$

We note that here we slightly abuse the notation of $\theta_j^*$ and $v_j^*$ in (3,4) and (6,7) for simplification. In the later context, $\theta_j^*$ and $v_j^*$ are used based on their definitions in (6,7).

**Connection to K-means** By viewing the target distribution as a set of particles that we want to cluster, in K-means clustering, each centroid (i.e., K-means center) represents one of the K disjoint groups[5] of particles, which is formed by assigning each particle in the whole set to the closest centroid among all the K centroids. K-means learns the optimal K centroids in the way that they can best approximate the whole set. The 'closeness' for assigning the particles is measured by the distance between the two points. As pointed out by Canas & Rosasco (2012), K-means essentially searches the optimal particle distribution formed by the K centroids that minimizes its Wasserstein distance to the target distribution. Our centroid approximation idea follows the same fashion of clustering but our key innovation is to measure the 'closeness' by examining the bootstrap loss of the centroids so that we can still learn the optimal centroids without obtaining the i.i.d. bootstrap particles first. We also point out that, while we share the same objective as K-means, the optimization algorithms differ. The Expectation-Maximization type of algorithm used by K-means is not applicable to our scenario.

**Comparing with Other Particle Improving Approach** Intuitively, from a high level abstracted perspective, we provide an approach to use K-means type of idea to improve the particle quality *without accessing to the true target distribution*. This is the key differentiator of this work to other approaches that improve the particle quality, as they all require to access the target distribution. For example Claici et al. (2018) requires that sampling from target distribution is cheap and easy. Chen et al. (2012; 2018a); Campbell & Beronov (2019) need to access the logarithm of the probability density function of the target distribution. In our problem, neither sampling from the target distribution is cheap nor the logarithm of the probability density function is available, making those approaches not more applicable.

### 3.1 TRAINING

The optimization of (6) can be solved by gradient descent. Suppose $\theta_j^*(t)$ is the $j$-th centroid at iteration $t$. We initialize $\{\theta_j^*(0)\}_{j=1}^m$ by sampling from $\rho_\pi$ and at iteration $t$, we update $\theta_t^*$ by applying the gradient descent on the loss in (6), which yields

$$\begin{aligned}\theta_j^*(t+1) &\leftarrow \theta_j^*(t) - \epsilon_t g(\theta_j^*(t)), \\ g(\theta_j^*(t)) &= \nabla_\theta \mathbb{E}_{\boldsymbol{w}\sim\pi}\left[\mathbb{I}\{j\in u_{\boldsymbol{w}}(t)\}\mathcal{L}_{\boldsymbol{w}}(\theta_j^*(t))\right]/v_j^*(t),\end{aligned} \tag{8}$$

where we define the index of the closest centroid to particle $\hat{\theta}_{\boldsymbol{w}}$ as $u_{\boldsymbol{w}}(t) = \arg\min_{j\in[m]}\mathcal{L}_{\boldsymbol{w}}(\theta_j^*(t))$ and $v_j^*(t) = \mathbb{P}_{\boldsymbol{w}\sim\pi}(j\in u_{\boldsymbol{w}}(t))$ denotes the probability that centroid $j$ is the one that gives the lowest bootstrap loss. The denominator $v_j^*(t)$ in $g(\theta_j^*(t))$ is optional. However, notice that the magnitude of numerator in $g(\theta_k^*(t))$ decays with larger $m$, which might require an adjustment of the learning rate when $m$ changes. This adjustment can be avoided by rescaling with $v_k^*(t)$.

We note that $\{\theta_j^*(0)\}_{j=1}^m$ is just $m$ i.i.d. bootstrap particles which is not optimal for approximation and our algorithm can be viewed as an approach for refining the $m$ particles by solving (6). In practice, we find that we can simply use random initialization (e.g., draw $\theta$ from some Gaussian distribution) instead.

**Centroid Degeneration Phenomenon** Naively applying the updating rule (8) may cause a degeneration phenomenon: When a centroid happens to give considerably worse performance than others, which can be caused by the stochasticity of gradient or worse initialization, the performance of this centroid will remain considerably worse throughout the optimization. The reason is simple. As this

---

[5]i.e. the regions separated by the dashed lines in the right plot of Figure 1.

centroid (e.g. $\theta_j^*(t)$) gives a considerably worse performance, the probability that it gives the lowest bootstrap loss, i.e., $v_j^*(t)$, is small. As a consequence, the gradient that updates this centroid is only based on aggregating information from a small low-density region of $\pi$ and hence can be unstable and further degrades this centroid. Note that this mechanism is self-reinforced since when this centroid cannot be effectively improved in the current iteration, it faces the same issue in the next one. As a result, this centroid is always significantly worse than the others.

We call this undesirable phenomenon *centroid degeneration* and we want to prevent this phenomenon because when it happens, we have a centroid that is not representative and contributes less to approximating $\rho_\pi$. We solve this issue with a simple solution and here is the intuition. The reason that a centroid degenerates lies in that this centroid is far from the good region where it gives a good performance. And when this happens, we should push the centroid to move towards this good region, which can be achieved by using the common gradient over the whole training data. Specifically, we define a threshold $\gamma$, indicating centroid $j$ is degenerated if $v_j^*(t) \leq \gamma$. And when it happens, we update using the common gradient over the whole data:

$$\theta_j^*(t+1) \leftarrow \theta_j^*(t) - \epsilon_t \nabla_\theta \mathcal{L}(\theta_j^*(t)). \tag{9}$$

In section 4, we give a theoretical analysis on why this modification is important and is able to solve the centroid degeneration issue.

**Practical Algorithm** In practice, we estimate the gradient by replacing the expectation over $\boldsymbol{w} \sim \pi$ in (8) with averaging over $M$ i.i.d. samples $\{\boldsymbol{w}_h\}_{h=1}^M$ drawn from $\pi$:

$$\hat{g}(\theta_j^*(t)) = \frac{\sum_{h=1}^M \left[ \mathbb{I}\{j \in u_{\boldsymbol{w}_h}(t)\} \nabla_\theta \mathcal{L}_{\boldsymbol{w}_h}(\theta_j^*(t)) \right]}{\sum_{h=1}^M \mathbb{I}\{j \in u_{\boldsymbol{w}_h}(t)\}}. \tag{10}$$

We emphasize that here $u_{\boldsymbol{w}_h}$ and $\mathcal{L}_{\boldsymbol{w}_h}$ for all $w_h$ can be computed very cheaply, enabling us to use a very large $M$ to reduce the error of gradient estimation. Specifically, at iteration $t$, for each $j \in [m]$, we first calculate

$$\mathbf{L}(\theta_j^*(t)) = [\ell(x_1, f_{\theta_j^*(t)}), ... \ell(x_n, f_{\theta_j^*(t)})]^\top \in \mathbb{R}^n, \tag{11}$$

which is the vector encodes the loss of centroid $j$ at each data point. This procedure does not introduce any extra computational overhead compared with standard gradient descent. After that, the bootstrap loss of centroid $j$ can be computed cheaply by $\mathcal{L}_{\boldsymbol{w}}(\theta_j^*(t)) = \boldsymbol{w}^\top \mathbf{L}(\theta_j^*(t))$. Similarly, it is cheap to obtain $u_{\boldsymbol{w}_h}$ by

$$u_{\boldsymbol{w}_h}(t) = \arg\min_{j \in [m]} \boldsymbol{w}_h^\top \mathbf{L}(\theta_j^*(t)). \tag{12}$$

Taking the modified updating rule introduced to prevent the centroid degeneration phenomenon into account, we update $\theta_j^*(t)$ by $\theta_j^*(t+1) \leftarrow \theta_j^*(t) - \epsilon_t \phi(\theta_j^*(t))$, where

$$\phi(\theta_j^*) = \begin{cases} \hat{g}(\theta_j^*(t)) & \text{if } \sum_{h \in [M]} \mathbb{I}\{u_{\boldsymbol{w}_h}(t) = j\}/M > \gamma \\ \nabla_\theta \mathcal{L}(\theta_j^*(t)) & \text{otherwise.} \end{cases} \tag{13}$$

Notice that as as $\mathbf{L}(\theta_j^*(t))$ is pre-computed, calculating $\mathcal{L}_{\boldsymbol{w}_h}(\theta_j^*(t))$ for many (e.g., $M$) different $\boldsymbol{w}_h$ is almost free, since it only requires a simple matrix multiplication with $O(nM)$ complexity. Similarly, calculating $u_{\boldsymbol{w}_h}(t)$ is also very cheap. In practical implementation, as $u_{\boldsymbol{w}_h}(t)$ do not change much within a few iterations, we can update $u_{\boldsymbol{w}_h}(t)$ every a few iterations (e.g., every epoch). We can also replace the $\nabla_\theta \mathcal{L}_{\boldsymbol{w}_h}(\theta_j^*(t))$ or $\nabla_\theta \mathcal{L}(\theta_j^*(t))$ in (13) using a mini-batch of data instead of the whole data, which leads to a stochastic gradient version of our algorithm. We refer readers to Algorithm 1 for the ideal updating and Algorithm 2 for the practical implementation in Appendix B for more details.

## 4 THEORY

Recall that, as discussed in (5), our approach relies on the intuition that bootstrap particles are nested in a small region so that we can approximate the distance between the centroid and a bootstrap particle by the bootstrap loss of that centroid. The main goal of this section is to give a formal theoretical justification of this intuition.

Before we proceed, we clarify several important setups for establishing and interpreting the theoretical result. As discussed in the introduction, we are mainly interested in the scenerio that the number of available particles/centroids $m$ is small while the number of training data $n$ is large, which motivates us to establish theoretical result in the region of small $m$ and large $n$. This is significantly different from conventional asymptotic analysis in which we aim to show the behavior when $m \to \infty$. We consider the setting that the parameter dimension $d$ is fixed and does not scale with $n$.

*We are mainly interested in characterizing the approximation of the proposed loss in (6) to the ideal loss in (3), given any small and fixed number $m$ of centroids when $n \to \infty$. This justifies why the proposed centroid approximation method can be viewed as minimizing the Wasserstein distance between the particle distribution $\rho_\pi^*$ and the target bootstrap distribution $\rho_\pi$.*

For simplicity, we build our analysis assuming the ideal update rule (8,9) is used. We start with the following main assumptions.

**Assumption 1 (Smoothness and boundedness)** *Assume that the following quantities are upper bounded by some constant $c < \infty$:*

1. $\max\limits_{i,j,k\in[d]} \sup\limits_{\theta\in\Theta, x\in\mathcal{X}} \dfrac{\partial^3 \ell(x, f_\theta)}{\partial_i \theta_i \partial \theta_j \partial \theta_k}$; 2. $\sup\limits_{\theta_1,\theta_2\in\Theta} \sup\limits_{x\in\mathcal{X}} \dfrac{\|\nabla_\theta^2 \ell(x, f_{\theta_1}) - \nabla_\theta^2 \ell(x, f_{\theta_2})\|}{\|\theta_1 - \theta_2\|}$;

3. $\sup\limits_{x\in\mathcal{X}, \theta\in\Theta} \left\|\nabla_\theta^2 \ell(x, f_\theta)\right\|$;       4. $\sup\limits_{\theta\in\Theta} \|\theta\|$.

Assumption 1 is a standard regularity condition on the boundness and smoothness of the problem.

**Assumption 2 (Asymptotic normality)** *Assume $\sqrt{n}\left(\hat{\theta}_w - \hat{\theta}\right) \xrightarrow{d} \mathcal{N}(0, A)$ and $\sqrt{n}\left(\hat{\theta} - \theta_0\right) \xrightarrow{d} \mathcal{N}(0, A)$ as $n \to \infty$, where $A$ is a positive-definite matrix with the largest eigenvalue bounded.*

Assumption 2 is a higher level assumption on the asymptotic normality of the estimators. Such result is classic and can be derived with some weak and technical regularity conditions. See examples in Chatterjee et al. (2005); Cheng et al. (2010).

**Assumption 3 (On the global minimizer)** *Suppose that $\lambda_{\min}\left(\nabla_\theta^2 \mathcal{L}_\infty(\theta_0)\right) > 0$.*

Assumption 3 is also standard showing the locally strongly convexity of the loss around the truth $\theta_0$.

**Assumption 4 (On the learning rate)** *Suppose that $\max_t \epsilon_t = O(n^{-1})$.*

Assumption 4 assumes that the learning rate of the algorithm is sufficiently small such that its induced discretization error is not the dominating term.

The key challenge of our analysis is to show that our dynamics is $\mathcal{B}(\theta_0, r)$-stable (defined below in Definition 1) for some small $r$, saying that $\{\theta_j^*(t)\}_{j=1}^m$ stay in a small region that is close to $\theta_0$ *for any iteration $t$*. Combined with the property[6] that $\hat{\theta}_w$ are also close to $\theta_0$, the centroids and the bootstrap particles are close to each other and thus our approximation in (5) holds for all $t \geq 0$. In this way, optimizing the centroids by minimizing our loss is almost equivalent to optimizing the centroids by minimizing the Wasserstein distance.

**Definition 1 ($\mathcal{B}(\theta, r)$-stable)** *Given some $\theta \in \Theta$ and $r \geq 0$, we say our dynamics is $\mathcal{B}(\theta, r)$-stable if $\forall t \geq 0$ and $\forall j \in [m]$, $\theta_j^*(t) \in \mathcal{B}(\theta, r)$, where $\mathcal{B}(\theta, r) := \{\theta' : \|\theta' - \theta\| \leq r, \theta' \in \Theta\}$ is the ball with radius $r$ centered at $\theta$.*

The key intuition to establish such $\mathcal{B}(\theta_0, r)$-stable result is to characterize that our optimization dynamics is implicitly *self-controlled*: when some centroid approaches the boundary of $\mathcal{B}(\theta_0, r)$, the updating mechanism automatically start to push the centroid to move towards the center of the region. Thus, if all the centroids are within $\mathcal{B}(\theta_0, r)$ at initialization, they will alway stay in this region.

Thanks to assumption 2, 3, when the dataset is large, the landscape of our loss is locally strongly convex around $\theta_0$. When a centroid $j$ is at the boundary of $\mathcal{B}(\theta_0, r)$, it has $v_j^*(t) < \gamma$ and thus the

---

[6]This is implied by the asymptotic normality in assumption 2.

|  |  |  | $m = 20$ | $m = 50$ | $m = 100$ | $m = 200$ |
|---|---|---|---|---|---|---|
| $\alpha = 0.9$ | Normal | Bootstrap | $0.029 \pm 0.010$ | $0.031 \pm 0.011$ | $0.021 \pm 0.010$ | $0.017 \pm 0.010$ |
|  |  | Centroid | $\mathbf{0.027 \pm 0.010}$ | $\mathbf{0.001 \pm 0.009}$ | $\mathbf{0.012 \pm 0.010}$ | $\mathbf{0.016 \pm 0.010}$ |
|  | Percentile | Bootstrap | $0.101 \pm 0.013$ | $0.036 \pm 0.011$ | $0.021 \pm 0.010$ | $\mathbf{0.014 \pm 0.010}$ |
|  |  | Centroid | $\mathbf{0.081 \pm 0.012}$ | $\mathbf{0.021 \pm 0.010}$ | $\mathbf{0.020 \pm 0.010}$ | $0.015 \pm 0.010$ |
|  | Pivotal | Bootstrap | $0.106 \pm 0.013$ | $0.045 \pm 0.011$ | $0.025 \pm 0.010$ | $0.023 \pm 0.010$ |
|  |  | Centroid | $\mathbf{0.046 \pm 0.011}$ | $\mathbf{0.013 \pm 0.009}$ | $\mathbf{0.011 \pm 0.010}$ | $\mathbf{0.020 \pm 0.010}$ |

Table 1: Centroid approximation for confidence interval. The numbers in the table represent $|\alpha - \hat{\alpha}|$, where $\hat{\alpha}$ is the estimated coverage probability. The errors bar is the standard deviation.

updating direction is the gradient of loss $\mathcal{L}$. By the convexity, such gradient will push the centroid move towards the center of $\mathcal{B}(\theta_0, r)$ where the empirical minimizer locates at. On the other hand, for centroid $j$ with $v_j^*(t) \geq \gamma$, its updating direction aggregates information from sufficient data point and thus behaves similarly to that of the common gradient, pushing centroid to move towards the center with the centroid is not close to the center.

**Theorem 1** *Under Assumptions 1-4 and suppose that we initialize $\theta_j^*(0)$, $j \in [m]$ by sampling from $\rho_\pi$, given any $m < \infty$ and $\gamma > 0$, when n is sufficiently large, we have*

$$\max_{j \in [m]} \sup_{t \geq 0} \left\| \theta_j^*(t) - \theta_0 \right\| = O_p(\sqrt{(\log n)/n}).$$

*Here the probability is taken w.r.t. training data.*

Theorem 1 implies our dynamics is $\mathcal{B}(\theta_0, r_n)$-stable with $r_n = O(\sqrt{\log n/n})$. The condition that $\theta_j^*(0) \sim \rho_\pi$ i.i.d. can be replaced by the condition that $\theta_j^*(0)$ is sufficiently close to $\theta_0$. We need such condition as we uniformly bound the distance between $\theta_j^*(t)$ and $\theta_0$ at any iteration including the first one. Theorem 1 implies that the approximation stated in (5) holds with high probability and hence the proposed loss in (6) is 'almost as good as' the ideal loss in (3).

**Theorem 2** *Under the same assumptions as Theorem 1, given any $m < \infty$ and $\gamma > 0$, when n is sufficiently large, we have*

$$\sup_{t \geq 0} \left| \mathbb{E}_{\boldsymbol{w} \sim \pi} \left[ \min_{j \in [m]} \mathcal{L}_{\boldsymbol{w}}(\theta_j^*(t)) \right] - B - \mathbb{E}_{\boldsymbol{w} \sim \pi} \left[ \min_{j \in [m]} ||\theta_j^*(t) - \hat{\theta}_{\boldsymbol{w}}||_D^2 \right]/2 \right| = O_p(\sqrt{(\log n)/n^{3/2}}).$$

*Here the probability is taken w.r.t. training data and B is some constant independent from $\theta_j^*(t)$ for any $t \geq 0$ and $j \in [m]$.*

**Asymptotics when $m$ also grows** Although our main interest is the asymptotics with a small, fixed $m$ and growing $n$, we discuss here on asymptotics when $m$ also grows. As shown in Section 3 and introduction, our method can be viewed as an 'approximated' K-means on the target distribution. From Theorem 5.2 in Canas & Rosasco (2012), the particle distribution formed by the optimal centroids learned by K-means gives improved $O(m^{-1/d})$ convergence to any general target distribution in terms of Wasserstein distance, where $d$ is data dimension. In comparison, the particle distribution of i.i.d. sample only gives $O(m^{-1/(2d+4)})$ from Theorem 5.1 in Canas & Rosasco (2012). This implies that our approach potentially also has such a rate improvement. Note that the results in Canas & Rosasco (2012) are for general target distribution without any $n$ involves. To rigorously establish the large $m$ asymptotic result for our problem, we need to study the joint limit of $n$ and $m$. This is indeed very non-trivial: as discussed in Weed et al. (2019) (i.e. Proposition 14), when $n \to \infty$, the target distribution $\rho_\pi$ becomes a sharp Gaussian and the convergence rate of i.i.d. bootstrap particles will gradually improve to $O(m^{-1/2})$ (in a way that depends on $n$). It implies that when $n \gg m \to \infty$, our improvement may become only constant level. We find establishing such a theory is out the scope of this conference paper and leave it as future work.

## 5 EXPERIMENT

As discussed in the introduction, our main goal is to improve the quality of the particle distribution when only a limited number of particles/centroids is allowed, so that we can use less particles at

|  |  | $m = 3$ | $m = 4$ | $m = 5$ | $m = 10$ |
|---|---|---|---|---|---|
| Mushroom | Bootstrap | $3282.1 \pm 72.82$ | $3307.9 \pm 69.2$ | $3311.6 \pm 79.3$ | $3397.4 \pm 51.4$ |
|  | Centroid | $\mathbf{3702.7 \pm 89.76}$ | $\mathbf{3723.1 \pm 78.7}$ | $\mathbf{3799.6 \pm 84.2}$ | $\mathbf{3796.9 \pm 36.1}$ |
| Statlog | Bootstrap | $1864.3 \pm 6.4$ | $1869.2 \pm 5.2$ | $1877.2 \pm 4.1$ | $1877.0 \pm 2.7$ |
|  | Centroid | $\mathbf{1893.6 \pm 6.0}$ | $\mathbf{1892.6 \pm 3.6}$ | $\mathbf{1891.3 \pm 3.5}$ | $\mathbf{1892.6 \pm 2.8}$ |
| Financial | Bootstrap | $2255.77 \pm 58.45$ | $2265.42 \pm 58.17$ | $2269.33 \pm 56.36$ | $2281.35 \pm 56.65$ |
|  | Centroid | $\mathbf{2313.29 \pm 56.45}$ | $\mathbf{2315.32 \pm 56.75}$ | $\mathbf{2323.88 \pm 56.73}$ | $\mathbf{2325.54 \pm 56.05}$ |

Table 2: Results on the contextual bandit experiment. The numbers in the table represent the averaged reward with its standard deviation.

deployment, which reduces the memory consumption and the computational cost for making prediction. Thus, our experiment design will be focusing on comparing the testing performance of vanilla bootstrap and our centroid based approach when the same and a small number of particles/centroids is used. We apply our method to four applications: confidence interval construction, bootstrap method in contextual bandit, bootstrapped deep Q-network and bagging. Due to space limit, we refer to Appendix C.4 for the bagging experiment, Appendix C.5 for ablation study on the importance of modifying the gradient to overcome the centroid degeneration phenomenon. Although we are less interested in the computational cost of training, as discussed in Section 3, our method actually only introduces a little training computation overhead, which is another advantage of our method. We draw analysis on this aspect in Appendix C.6.

## 5.1 BOOTSTRAP CONFIDENCE INTERVAL

We start with a classic application of bootstrap: confidence interval estimation for linear model with parameter $\theta$. Fix confidence level $\alpha$, we consider three ways to construct (two-sided) bootstrap confidence interval of $\theta$: the Normal interval, the percentile interval and the pivotal interval. And we test $m = 20, 50, 100, 200$. For all experiments, we repeat with 1000 independent random trials. We consider the standard bootstrap as baseline. Detailed experimental setup are included in Appendix C.1.

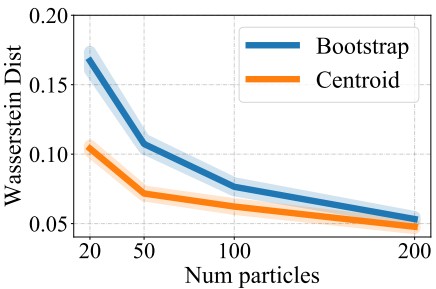

Figure 2: Wasserstein distance between the particle distribution and the true bootstrap distribution w.r.t. the number of particles.

Figure 2 shows the Wasserstein distance between the true target distribution $\rho_\pi$ and the empirical distributions obtained by (a) i.i.d. sampling $\hat{\rho}_\pi$, (b) the proposed centroid approximation $\rho_\pi^*$. The centroid approximation significantly reduces the Wasserstein distance by a large margin. We then compare the quality of obtained confidence intervals, which is measured by the difference between the estimated coverage probability and the true confidence level, i.e., $|\hat{\alpha} - \alpha|$ (the lower the better). Here we only consider confidence intervals of the first coordinate of $\theta$: $\theta_1$. Table 1 summarizes the result with $\alpha = 0.9$. We see that using more particles is generally able to improve the constructed confidence intervals. We also compare with two variants of standard bootstrap: Bayesian bootstrap (Rubin, 1981) and residual bootstrap (Efron, 1992). And we consider varying $\alpha = 0.8, 0.95$. These results are included in Appendix C.1.

## 5.2 CENTROID APPROXIMATION FOR BOOTSTRAP METHOD IN CONTEXTUAL BANDIT

Contextual bandit is a classic task in sequential decision making, in which accurately quantifying the model uncertainty is important in order to achieve good exploration-exploitation trade-off. As shown in Riquelme et al. (2018), tracking the model uncertainty using bootstrap is a strong method for contextual bandit. However, it is costly to maintain a large number of bootstrap models and thus the number of models is typically within 10 (Osband et al., 2016). We find that applying the proposed centroid approximation here can significantly improve the performance. Riquelme et al. (2018) uses $m = 3$ bootstrap models and we give a more comprehensive evaluation with $m = 3, 4, 5, 10$. We consider three datasets: Mushroom, Statlog and Financial. We set $\gamma = 0.5/m$. We randomly generate 20 different context sequences, apply all the methods and report the averaged cumulative reward and its standard deviation. Table 2 summarizes the result and note that a large part of variance

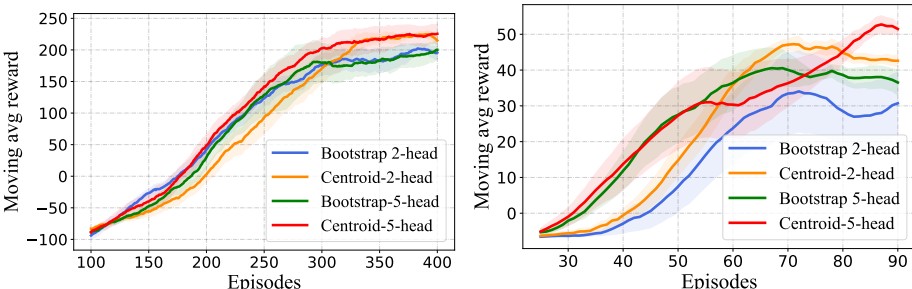

Figure 3: Results for Bootstrap DQN with centroid approximation experiment. Left: LunarLander-v2; Right: Catcher-v0.

can be explained by different context sequences. All results in Table 2 are statistically significant under significant level 5% using matched pair t-test. Table 2 shows that using more bootstrap models generally improves the accumulated reward. And when using the same number of models, the proposed centroid approximation method consistently improves over standard bootstrap method. We refer readers to appendix C.2 for more information on the background and experiment.

## 5.3 CENTROID APPROXIMATION FOR BOOTSTRAP DQN

"Efficient exploration is a major challenge for reinforcement learning (RL). Common dithering strategies such as $\epsilon$-greedy do not carry out temporally-extended exploration, which leads to exponentially larger data requirements" (Osband et al., 2016). To tackle this issue, Osband et al. (2016) proposes the Bootstrapped Deep Q-Network (DQN). We apply our centroid approximation to improve Boot-strapped DQN. We consider $m = 2$, 5 and similar to the experimental setting in contextual bandit, we set $\gamma = 0.5/m$. We consider two benchmark environments: LunarLander-v2 and Catcher-v0 from GYM (Brockman et al., 2016) and PyGame learning environment (Tasfi, 2016). We conduct the experiment with 5 independent random trails and report the averaged result with its standard deviation. We refer readers to Appendix C.3 for more background and other experiment details. Figure 3 summarizes the result. For LunarLander-v2, Bootstrap DQN with 2 and 5 heads give similar performance but both converge to a less optimal model compared with the centroid approximation method. Centroid approximation method with 2 and 5 heads performs similarly at convergence but centroid approximation method with 5 heads is able to converge faster than 2-head model and thus has lower regret. For Catcher-v0, adding more heads to the model is able to improve the performance for both methods. The proposed centroid approximation consistently improves over baselines.

## 6 RELATED WORK

**Bootstrap** is an classical statistical inference method, which was developed by Efron (1992) and generalized by, i.e., Mammen (1993); Shao (2010); Efron (2012) (just to name a few). Bootstrap can be widely applied to various statistical inference problem, such as confidence interval estimation (DiCiccio et al., 1996), model selection (Shao, 1996), high-dimensional inference (Chen et al., 2018b; El Karoui & Purdom, 2018; Nie & Ročková, 2020), off-policy evaluation (Hanna et al., 2017), distributed inference (Yu et al., 2020) and inference for ensemble model (Kim et al., 2020), etc.

Despite its wide applications and nice theoretical properties, there has been very few works on discussing and improving the approximation efficiency *in the region of small bootstrap sample size*, beyond the i.i.d. sampling paradigm. While the m-out-of-n bootstrap (Bickel et al., 2012) and the bag of little bootstrap (Kleiner et al., 2014) are designed to reduce the computational cost with the subsampling techniques in the big data settings (large $n$), they still require a large bootstrap sample size and thus are still not scalable to large deep learning applications.

**Bayesian Inference** is a different approach to quantify the model uncertainty. Different from frequentists' method, Bayesian assumes a prior over the model and the uncertainty can be captured by the posterior. Bayesian inference have been largely popularized in machine learning, largely thanks to the recent development in scalable sampling method (Welling & Teh, 2011; Chen et al., 2014; Seita et al., 2018; Wu et al., 2020), variational inference (Blei et al., 2017; Liu & Wang, 2016), and other approximation methods such as Gal & Ghahramani (2016); Lee et al. (2018). In comparison,

bootstrap has been much less widely used in modern machine learning and deep learning. We believe this is largely attributed to *the lack of similarly efficient computational methods in the small sample size $m$ region*, which is the very problem that we aim to address with our new centroid approximation method.

**Uncertainty in Deep Learning** In additional to the applications considered in this paper, uncertainty in deep learning model can also be applied to problems including calibration (Guo et al., 2017) and out-of-distribution detection (Nguyen et al., 2015). The definition of uncertainty of neural network is quite generalized (e.g., Gal & Ghahramani (2016); Ovadia et al. (2019); Maddox et al. (2019); Van Amersfoort et al. (2020)) and can be quite different from the uncertainty that bootstrap inference want to quantify and can be approached with various methods including drop out (Gal & Ghahramani, 2016; Durasov et al., 2020), label smoothing (Qin et al., 2020), designing new modules in the model (Kivaranovic et al., 2020), adversarial training (Lakshminarayanan et al., 2017) and Bayesian modeling (Blundell et al., 2015), etc. This paper focuses on improving the bootstrap method and thus is orthogonal to those previous works. Pearce et al. (2018); Salem et al. (2020) also try to refine the ensemble models to improve the quality of prediction interval. Compare with our method, their method can only be applied to prediction interval and does not have theoretical guarantee.

## 7 CONCLUSION

We propose a centroid approximation method to learn an improved particle distribution that better approximates the target bootstrap distribution, especially in the region with small particle size. Theoretically, when the size of training data is large, our objective function is surrogate to the Wasserstein distance between the particle distribution and target distribution. Thus, compared with standard bootstrap, the proposed centroid approximation method actively optimizes the distance between particle distribution and target distribution. The proposed method is simple and can be flexibly used for applications of bootstrap with negligible extra computational cost.

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

# A   PROOF

We also show Theorem 3, which gives a formal characterization of the Taylor approximation intuition introduced in (5).

**Theorem 3** *Under Assumption 1 and 2, when $n$ is sufficiently large, we have*

$$\mathcal{L}_{\boldsymbol{w}}(\theta) = \mathcal{L}_{\boldsymbol{w}}(\hat{\theta}_{\boldsymbol{w}}) + \tfrac{1}{2}\left(\theta - \hat{\theta}_{\boldsymbol{w}}\right)^{\top}\nabla_{\theta}^2\mathcal{L}_{\infty}(\theta_0)\left(\theta - \hat{\theta}_{\boldsymbol{w}}\right)$$
$$+ O_p\left(||\theta - \hat{\theta}_{\boldsymbol{w}}||^2(n^{-1/2} + ||\theta - \hat{\theta}_{\boldsymbol{w}}||)\right).$$

*Here the stochastic boundedness is taken w.r.t. the training data and $\boldsymbol{w}$.*

In the proof, we may use $c$ to represent some absolute constant, which may vary in different lines.

## A.1   PROOF OF THEOREM 3

With the fact that $\nabla_{\boldsymbol{w}}\mathcal{L}(\hat{\theta}_{\boldsymbol{w}}) = 0$ and under assumption 1, using Taylor expansion, we have

$$\mathcal{L}_{\boldsymbol{w}}(\theta) = \mathcal{L}_{\boldsymbol{w}}(\hat{\theta}_{\boldsymbol{w}}) + \frac{1}{2}\left(\theta - \hat{\theta}_{\boldsymbol{w}}\right)^{\top}\nabla_{\theta}^2\mathcal{L}_{\boldsymbol{w}}(\hat{\theta}_{\boldsymbol{w}})\left(\theta - \hat{\theta}_{\boldsymbol{w}}\right) + O\left(\left\|\theta - \hat{\theta}_{\boldsymbol{w}}\right\|^3\right).$$

Notice that

$$\left|\left(\theta - \hat{\theta}_{\boldsymbol{w}}\right)^{\top}\left(\nabla_{\theta}^2\mathcal{L}_{\boldsymbol{w}}(\hat{\theta}_{\boldsymbol{w}}) - \nabla_{\theta}^2\mathcal{L}_{\infty}(\theta_0)\right)\left(\theta - \hat{\theta}_{\boldsymbol{w}}\right)\right|$$

$$\leq \left\|\theta - \hat{\theta}_{\boldsymbol{w}}\right\|^2\left\|\nabla_{\theta}^2\mathcal{L}_{\boldsymbol{w}}(\hat{\theta}_{\boldsymbol{w}}) - \nabla_{\theta}^2\mathcal{L}_{\infty}(\theta_0)\right\|$$

$$\leq \left\|\theta - \hat{\theta}_{\boldsymbol{w}}\right\|^2\left(\left\|\nabla_{\theta}^2\mathcal{L}_{\boldsymbol{w}}(\hat{\theta}_{\boldsymbol{w}}) - \nabla_{\theta}^2\mathcal{L}_{\boldsymbol{w}}(\theta_0)\right\| + \left\|\nabla_{\theta}^2\mathcal{L}_{\boldsymbol{w}}(\theta_0) - \nabla_{\theta}^2\mathcal{L}_{\infty}(\theta_0)\right\|\right)$$

$$\leq \left\|\theta - \hat{\theta}_{\boldsymbol{w}}\right\|^2\left(C\left\|\hat{\theta}_{\boldsymbol{w}} - \theta_0\right\| + \left\|\nabla_{\theta}^2\mathcal{L}_{\boldsymbol{w}}(\theta_0) - \nabla_{\theta}^2\mathcal{L}_{\infty}(\theta_0)\right\|_F\right),$$

where we denote the Frobenius norm as $\|\cdot\|_F$. With assumption 2, we have $\left\|\hat{\theta}_{\boldsymbol{w}} - \theta_0\right\| = O_p(n^{-1/2})$. By applying centroid limit theorem and delta method to $\left|\nabla_{\theta_{ij}}^2\mathcal{L}_{\boldsymbol{w}}(\theta_0) - \nabla_{\theta_{ij}}^2\mathcal{L}_{\infty}(\theta_0)\right|$ for every pair $i, j \in [d]$, we have $\left\|\nabla_{\theta}^2\mathcal{L}_{\boldsymbol{w}}(\theta_0) - \nabla_{\theta}^2\mathcal{L}_{\infty}(\theta_0)\right\|_F = O_p(n^{-1/2})$. Thus we obtained the desired result.

## A.2   PROOF OF THEOREM 1

Given any radius $r$ and $\epsilon > 0$, with sufficiently large $n$, we have

$$\mathbb{P}\left(\left\|\hat{\theta}_w - \theta_0\right\| \geq r\right) \leq \exp(-\lambda n r^2) + \epsilon/m,$$

for $\lambda = \frac{1}{4}\lambda_{\max}(A)^{-1}$. Here the probability is the jointly probability of bootstrap weight and training data. Thus, given any $r$, under the assumption that $\theta_j^*(0)$ is initialized via sampling $\hat{\theta}_w$, then we have

$$\mathbb{P}\left(\cup_{j\in[m]}\left\{\left\|\theta_j^*(0) - \theta_0\right\| \geq r\right\}\right) \leq \sum_{j\in[m]}\mathbb{P}\left(\left\|\theta_j^*(0) - \theta_0\right\| \geq r\right) \leq m\exp(-\lambda n r^2) + \epsilon.$$

We proof by induction. Given any $\{\theta_j\}_{j=1}^m$, define

$$R_{k,r} = \mathbb{I}\left\{w \in \text{supp}(\pi) : \arg\min_{j\in[m]}\mathcal{L}_{\boldsymbol{w}}(\theta_j) = k \text{ and } \left\|\hat{\theta}_w - \theta_0\right\| \leq r\right\}.$$

Suppose at iteration $t$, we have $\|\theta_k^*(t) - \theta_0\| \leq \frac{c\alpha\sqrt{\frac{\log n}{n}}}{\lambda_0\gamma}$ for some constant $c$ and $\lambda_0$, which we denote as the minimum eigenvalue of $\nabla_{\theta}^2\mathcal{L}_{\infty}(f_{\theta_0})$. Now at iteration $t$, we have two cases.

**Case 1:** $\mathbb{E}_\pi R_{k,\infty} \geq \gamma$ Suppose that at iteration $t$, for $k$ such that $\mathbb{E}_\pi R_{k,\infty} \geq \gamma$, and $\|\theta_k^*(t) - \theta_0\| = q_k$, we have the following property:

$$\|\theta_k^*(t+1) - \theta_0\|^2 = \left\|\theta_k^*(t) - \frac{\epsilon_t}{\mathbb{E}_\pi R_{k,\infty}} \mathbb{E}_\pi \left[\nabla_\theta \mathcal{L}_{\boldsymbol{w}}(\theta_k^*(t)) R_{k,\infty}\right] - \theta_0\right\|^2$$

$$= \|\theta_k^*(t) - \theta_0\|^2 - \frac{2\epsilon_t}{\mathbb{E}_\pi R_{k,\infty}} \langle \theta_k^*(t) - \theta_0, \mathbb{E}_\pi \left[\nabla_\theta \mathcal{L}_{\boldsymbol{w}}(\theta_k^*(t)) R_{k,\infty}\right]\rangle + \epsilon_t^2 \|\mathbb{E}_\pi \left[\nabla_\theta \mathcal{L}_{\boldsymbol{w}}(\theta_k^*(t)) R_{k,\infty}\right]\|^2.$$

Notice that

$$\mathbb{E}_\pi \left[\nabla_\theta \mathcal{L}_{\boldsymbol{w}}(\theta_k^*(t)) R_{k,q_k}\right] \stackrel{(1)}{=} \mathbb{E}_\pi \left[\nabla_\theta^2 \mathcal{L}_{\boldsymbol{w}}(\hat{\theta}_w)(\theta_k^*(t) - \hat{\theta}_w) R_{k,q_k}\right] + o\left(q_k^2\right)$$

$$\stackrel{(2)}{=} \mathbb{E}_\pi \left[\nabla_\theta^2 \mathcal{L}_{\boldsymbol{w}}(\hat{\theta}_w)(\theta_k^*(t) - \theta_0) R_{k,q_k}\right] + O\left(q_k^2\right)$$

$$\stackrel{(3)}{=} \mathbb{E}_\pi \left[\nabla_\theta^2 \mathcal{L}_{\boldsymbol{w}}(\theta_0)(\theta_k^*(t) - \theta_0) R_{k,q_k}\right] + O\left(q_k^2\right)$$

Here (1) is obtained via applying Taylor expansion on $\nabla_\theta \mathcal{L}_{\boldsymbol{w}}(\theta_k^*(t))$ at $\hat{\theta}_{\boldsymbol{w}}$. (2) is by assumption 1 and 2. (3) is by assumption 1. We thus have

$$-\langle \theta_k^*(t) - \theta_0, \mathbb{E}_\pi \left[\nabla_\theta \mathcal{L}_{\boldsymbol{w}}(\theta_k^*(t)) R_{k,\infty}\right]\rangle$$

$$\leq -\langle \theta_k^*(t) - \theta_0, \mathbb{E}_\pi \left[\nabla_\theta \mathcal{L}_{\boldsymbol{w}}(\theta_k^*(t)) R_{k,q_k}\right]\rangle + \|\theta_k^*(t) - \theta_0\| \|\mathbb{E}_\pi \nabla_\theta \mathcal{L}_{\boldsymbol{w}}(\theta_k^*(t))(1 - R_{k,q_k})\|$$

$$\leq -\langle \theta_k^*(t) - \theta_0, \mathbb{E}_\pi \left[\nabla_\theta \mathcal{L}_{\boldsymbol{w}}(\theta_k^*(t)) R_{k,q_k}\right]\rangle + cq_k \exp(-\lambda n q_k^2)$$

$$\leq -\mathbb{E}_\pi R_{k,q_k} (\theta_k^*(t) - \theta_0)^\top \nabla_\theta^2 \mathcal{L}_{\boldsymbol{w}}(\theta_0)(\theta_k^*(t) - \theta_0) + cq_k \exp(-\lambda n q_k^2) + O\left(q_k^3\right).$$

Notice that with sufficiently large $n$, with central limit theorem, we have

$$-\mathbb{E}_\pi R_{k,q_k}(\theta_k^*(t) - \theta_0)^\top \nabla_\theta^2 \mathcal{L}_{\boldsymbol{w}}(\theta_0)(\theta_k^*(t) - \theta_0)$$

$$\leq \|\theta_k^*(t) - \theta_0\|^2 \mathbb{E}_\pi \left\|\nabla_\theta^2 \mathcal{L}_{\boldsymbol{w}}(\theta_0) - \nabla_\theta^2 \mathcal{L}_{\boldsymbol{w}}(\theta_0)\right\| - \mathbb{E}_\pi R_{k,q_k}(\theta_k^*(t) - \theta_0)^\top \nabla_\theta^2 \mathcal{L}_\infty(\theta_0)(\theta_k^*(t) - \theta_0)$$

$$= -\mathbb{E}_\pi R_{k,q_k}(\theta_k^*(t) - \theta_0)^\top \nabla_\theta^2 \mathcal{L}_\infty(\theta_0)(\theta_k^*(t) - \theta_0) + O(n^{-1/2}).$$

This gives that

$$-\langle \theta_k^*(t) - \theta_0, \mathbb{E}_\pi \left[\nabla_\theta \mathcal{L}_{\boldsymbol{w}}(\theta_k^*(t)) R_{k,\infty}\right]\rangle$$

$$\leq -\mathbb{E}_\pi R_{k,q_k}(\theta_k^*(t) - \theta_0)^\top \nabla_\theta^2 \mathcal{L}_\infty(\theta_0)(\theta_k^*(t) - \theta_0) + cq_k \exp(-\lambda n q_k^2) + O\left(q_k^3 + q_k n^{-1/2}\right)$$

$$\leq -\lambda_0 \mathbb{E}_\pi R_{k,q_k} \|\theta_k^*(t) - \theta_0\|^2 + cq_k \exp(-\lambda n q_k^2) + O\left(q_k^3 + q_k n^{-1/2}\right).$$

Use the above estimation, we have

$$\|\theta_k^*(t+1) - \theta_0\|^2$$

$$\leq \|\theta_k^*(t) - \theta_0\|^2 - 2\epsilon_t \lambda_{\min} \frac{\mathbb{E}_\pi R_{k,q_k}}{\mathbb{E}_\pi R_{k,\infty}} \|\theta_k^*(t) - \theta_0\|^2$$

$$+ 2c\epsilon_t q_k \exp(-\lambda n q_k^2)/\mathbb{E}_\pi R_{k,\infty} + O\left(\epsilon_t(q_k^3 + q_k n^{-1/2})/\mathbb{E}_\pi R_{k,\infty} + \epsilon_t^2\right)$$

$$\leq \|\theta_k^*(t) - \theta_0\|^2 + \frac{\epsilon_t}{\mathbb{E}_\pi R_{k,\infty}} \left(-2\lambda_0 \mathbb{E}_\pi R_{k,q_k} \|\theta_k^*(t) - \theta_0\|^2 - 2cq_k \exp(-\lambda n q_k^2) + O\left(q_k^3 + \epsilon_t + q_k n^{-1/2}\right)\right).$$

Notice that by choosing $\alpha > \sqrt{1/(2\lambda)}$ and $\epsilon_t = O(n^{-1})$, with sufficiently large $n$, when

$$\|\theta_k^*(t) - \theta_0\| \geq \frac{c\alpha \sqrt{\frac{\log n}{n}}}{\lambda_0 \mathbb{E}_\pi R_{k,q_k}}$$

for some constant $c$, we have

$$\|\theta_k^*(t+1) - \theta_0\| \leq \|\theta_k^*(t) - \theta_0\|.$$

Thus $\|\theta_k^*(t+1) - \theta_0\| \leq \frac{c\alpha \sqrt{\frac{\log n}{n}}}{\lambda_0 \mathbb{E}_\pi R_{k,\infty}} \leq \frac{c\alpha \sqrt{\frac{\log n}{n}}}{\lambda_0 \gamma}$ for some constant $c$.

**Case 2:** $\mathbb{E}_\pi R_{k,\infty} \le \gamma$    In this case, we have

$$\|\theta_k^*(t+1) - \theta_0\|^2 = \left\|\theta_k^*(t) - \epsilon_t \nabla_\theta \mathcal{L}(f_{\theta_k^*(t)}) - \theta_0\right\|^2$$

$$= \|\theta_k^*(t) - \theta_0\|^2 - 2\epsilon_t \langle\theta_k^*(t) - \theta_0, \nabla_\theta \mathcal{L}(\theta_k^*(t))\rangle + \epsilon_t^2 \left\|\nabla_\theta \mathcal{L}(f_{\theta_k^*(t)})\right\|^2.$$

Notice that

$$-\langle\theta_k^*(t) - \theta_0, \nabla_\theta \mathcal{L}(\theta_k^*(t))\rangle \le -\langle\theta_k^*(t) - \theta_0, \nabla_\theta^2 \mathcal{L}(\theta_0)(\theta_k^*(t) - \theta_0)\rangle - \langle\theta_k^*(t) - \theta_0, \nabla_\theta \mathcal{L}(f_{\theta_0})\rangle + o(\|\theta_k^*(t) - \theta_0\|^3)$$

$$= -(\theta_k^*(t) - \theta_0)^\top \nabla_\theta^2 \mathcal{L}_\infty(\theta_0)(\theta_k^*(t) - \theta_0) + o(\|\theta_k^*(t) - \theta_0\|^3) + O_p(n^{-1/2})\|\theta_k^*(t) - \theta_0\|.$$

This gives that

$$\|\theta_k^*(t+1) - \theta_0\|^2 \le \|\theta_k^*(t) - \theta_0\|^2 - 2\epsilon_t \lambda_0 \|\theta_k^*(t) - \theta_0\|^2 + o(\epsilon_t \|\theta_k^*(t) - \theta_0\|^3 + \epsilon_t^2) + O_p(n^{-1/2})\epsilon_t \|\theta_k^*(t) - \theta_0\|.$$

With $\epsilon_t = O(n^{-1})$ and sufficiently large $n$, when

$$\|\theta_k^*(t) - \theta_0\| \ge \frac{c\alpha\sqrt{\frac{\log n}{n}}}{\lambda_0 \gamma},$$

we have $\|\theta_k^*(t+1) - \theta_0\|^2 \le \|\theta_k^*(t) - \theta_0\|^2$.

Combine this two cases, we conclude that $\|\theta_k^*(t+1) - \theta_0\| \le \frac{c\alpha\sqrt{\frac{\log n}{n}}}{\lambda_0 \gamma}$ for any $t$, when $\|\theta_k^*(0) - \theta_0\| \le \frac{c\alpha\sqrt{\frac{\log n}{n}}}{\lambda_0 \gamma}$. We thus conclude that, for any $\alpha > \sqrt{1/(2\lambda)}$ and $\epsilon > 0$, when $n$ is sufficiently large, with probability at least $1 - m\exp\left(-\lambda\frac{c\alpha^2 \log n}{\lambda_0^2 \gamma^2}\right) - \epsilon$, we have

$$\max_{j \in [m]} \sup_t \left\|\theta_j^*(t) - \theta_0\right\| \le \frac{c\alpha\sqrt{\frac{\log n}{n}}}{\lambda_0 \gamma}.$$

### A.3   PROOF FOR THEOREM 2

Notice that

$$\mathcal{L}_w(\theta_j^*(t)) - \mathcal{L}_w(\hat{\theta}_w) = \frac{1}{2}\left(\theta_j^*(t) - \hat{\theta}_w\right)^\top \nabla_\theta^2 \mathcal{L}_w(\hat{\theta}_w)\left(\theta_j^*(t) - \hat{\theta}_w\right) + O(\left\|\theta_j^*(t) - \hat{\theta}_w\right\|^3)$$

$$= \frac{1}{2}\left(\theta_j^*(t) - \hat{\theta}_w\right)^\top \nabla_\theta^2 \mathcal{L}_w(\theta_0)\left(\theta_j^*(t) - \hat{\theta}_w\right) + O(\left\|\theta_j^*(t) - \hat{\theta}_w\right\|^3) + O(\left\|\theta_j^*(t) - \hat{\theta}_w\right\|^2 \left\|\hat{\theta}_w - \theta_0\right\|)$$

$$= \frac{1}{2}\left\|\theta_j^*(t) - \hat{\theta}_w\right\|_D^2 + \left\|\nabla_\theta^2 \mathcal{L}_w(\theta_0) - \nabla_\theta^2 \mathcal{L}_\infty(\theta_0)\right\|\left\|\theta_j^*(t) - \hat{\theta}_w\right\|^2 + O(\left\|\theta_j^*(t) - \hat{\theta}_w\right\|^3)$$

$$+ O(\left\|\theta_j^*(t) - \hat{\theta}_w\right\|^2 \left\|\hat{\theta}_w - \theta_0\right\|)$$

Given $w$, we define $u_w = \arg\min_{j \in [m]} \left\|\theta_j^*(t) - \hat{\theta}_w\right\|_D^2$. For any $\alpha > \sqrt{1/(2\lambda)}$ and $\epsilon > 0$, when $n$ is sufficiently large, with probability at least $1 - m\exp\left(-\lambda\frac{c\alpha^2 \log n}{\lambda_0^2 \gamma^2}\right) - \epsilon$, we have

$$\frac{1}{2}\mathbb{E}_{w \sim \pi}\left[\min_{j \in [m]}\left\|\theta_j^*(t) - \hat{\theta}_w\right\|_D^2\right]$$

$$= \frac{1}{2}\mathbb{E}_{w \sim \pi}\left[\left\|\theta_{u_w}^* - \hat{\theta}_w\right\|_D^2\right]$$

$$\ge \mathbb{E}_{w \sim \pi}\left[\mathcal{L}_w(\theta_{u_w}^*) - \mathcal{L}_w(\hat{\theta}_w) - c\left\|\theta_{u_w}^* - \hat{\theta}_w\right\|^2\left(\left\|\hat{\theta}_w - \theta_0\right\| + \left\|\nabla_\theta^2 \mathcal{L}_w(\theta_0) - \nabla_\theta^2 \mathcal{L}_\infty(\theta_0)\right\| + \left\|\theta_{u_w}^* - \hat{\theta}_w\right\|\right)\right]$$

$$\ge \mathbb{E}_{w \sim \pi}\left[\mathcal{L}_w(\theta_{u_w}^*)\right] - \mathbb{E}_{w \sim \pi}\left[\mathcal{L}_w(\hat{\theta}_w)\right] - c\left(\frac{\alpha\sqrt{\log n}}{\lambda_0 \gamma n^{3/2}}\right)$$

$$= \mathbb{E}_{w \sim \pi}\left[\min_{j \in [m]} \mathcal{L}_w(\theta_j^*(t))\right] - \mathbb{E}_{w \sim \pi}\left[\mathcal{L}_w(\hat{\theta}_w)\right] - c\left(\frac{\alpha\sqrt{\log n}}{\lambda_0 \gamma n^{3/2}}\right).$$

---

**Algorithm 1** Ideal algorithm for centroid approximation with full-batch gradient used and $w_h$ updated every iteration.

---

1: Initialize $\theta_j^*(0)$, $j \in [m]$ by i.i.d. sampling from $\rho_\pi$ or other distribution such as Gaussian.
2: **for** $t \in$ iterations **do**
3:     $\forall j \in [m]$, calculate $\mathbf{L}(\theta_j^*(t))$ defined in (11)
4:     Sample $\{w_h\}_{h=1}^M$, i.i.d. from $\pi$.
5:     $\forall h \in [M]$ and $j \in [m]$, calculate $\mathbf{L}_{w_h}(\theta_j^*(t)) = w_h^T \mathbf{L}(\theta_j^*(t))$.
6:     $\forall h \in [M]$, calculate $u_{w_h}$ defined in (12) for each $h$.
7:     $\forall j \in [m]$, update $\theta_j^*$ by (13).
8: **end for**

---

Similarly, we also have, with probability at least $1 - m \exp\left(-\lambda \frac{c\alpha^2 \log n}{\lambda_0^2 \gamma^2}\right)$,

$$\mathbb{E}_{w \sim \pi}\left[\min_{j \in [m]} \mathcal{L}_w(\theta_j^*(t))\right] - \mathbb{E}_{w \sim \pi}\left[\mathcal{L}_w(\hat{\theta}_w)\right] \geq \frac{1}{2}\mathbb{E}_{w \sim \pi}\left[\min_{j \in [m]} \left\|\theta_j^*(t) - \hat{\theta}_w\right\|_D^2\right] - c\left(\frac{\alpha\sqrt{\log n}}{\lambda_0 \gamma n^{3/2}}\right).$$

Notice that the above bound holds uniformly for all $j \in [m]$ and any iteration $t$, which implies that with probability at least $1 - 2m \exp\left(-\lambda \frac{c\alpha^2 \log n}{\lambda_0^2 \gamma^2}\right) - 2\epsilon$, we have

$$\sup_{t \geq 0}\left|\mathbb{E}_{w \sim \pi}[\min_{j \in [m]} \mathcal{L}_w(\theta_j^*(t))] - B - \mathbb{E}_{w \sim \pi}[\min_{j \in [m]} ||\theta_j^*(t) - \hat{\theta}_w||_D^2]/2\right| \leq c\left(\frac{\alpha\sqrt{\log n}}{\lambda_0 \gamma n^{3/2}}\right).$$

# B    ALGORITHM BOX

We provide pseudo algorithm for the ideal centroid approximation algorithm in Algorithm 1. In practical implementation, we do not need to update $w_h$ every iteration and can also replace the full-batch gradient by stochastic gradient. Specifically, notice that $g(\hat{\theta}_j^*)$ defined in (10) can be alternative represented as

$$\hat{g}(\theta_j^*(t)) = \frac{\sum_{h=1}^M \sum_{i=1}^n \left[\mathbb{I}\{j \in u_{w_h}(t)\}\right] w_{h,i} \nabla_\theta \ell(x_i, f_{\theta_j^*(t)})/n}{\sum_{h=1}^M \left[\mathbb{I}\{j \in u_{w_h}\}\right]} = \frac{1}{n}\sum_{i=1}^n q_{i,j}\nabla_\theta \ell(x_i, f_{\theta_j^*(t)}),$$

(14)

where $q_{i,j}$ is defined by

$$q_{ij} := \frac{\sum_{h=1}^M \sum_{i=1}^n \left[\mathbb{I}\{j \in u_{w_h}(t)\}\right] w_{h,i}}{\sum_{h=1}^M \left[\mathbb{I}\{j \in u_{w_h}(t)\}\right]}.$$

(15)

This allows us to use a stochastic gradient version of gradient

$$\hat{g}_{sgd}(\theta_j^*) = \frac{1}{|B|}\sum_{i \in [B]} q_{i,j}\nabla_\theta \ell(x_i, f_{\theta_j^*}),$$

(16)

where $B$ is the set of mini-batch data. The detailed algorithm is summarized in Algorithm

# C    ADDITIONAL EXPERIMENT DETAILS

## C.1    BOOTSTRAP CONFIDENCE INTERVAL

Given a model $f_\theta$ parameterized by $\theta$ and a training set with $n$ data points i.i.d. sampled from population, our goal is to construct confidence interval for $\theta$. Let $\tilde{\rho}_\pi$ be an empirical distribution approximating $\rho_\pi$, which could be obtained by i.i.d. sampling, or by our centroid method. Denote by $Q[\alpha, \tilde{\rho}_\pi]$ the $\alpha$-quantile function of $\tilde{\rho}_\pi$ with some $\alpha \in [0, 1]$. We consider the following three ways to construct (two-sided) bootstrap confidence interval of $\theta$ with confidence level $\alpha$: the Normal interval, the percentile interval and the pivotal interval which are defined below.

---

**Algorithm 2** Practical implementation of centroid approximation with less frequent updating of $\boldsymbol{w}_h$ and stochastic gradient enabled.

1: Initialize $\theta_j^*(0)$, $j \in [m]$ by i.i.d. sampling from $\rho_\pi$ or other distribution such as Gaussian.
2: **for** $t \in$ iterations **do**
3:  // Update $\boldsymbol{w}_h$ only every a few iterations.
4:  **if** $t$ mod freq $== 0$ **then**
5:    $\forall j \in [m]$, calculate $\mathbf{L}(\theta_j^*(t))$ defined in (11)
6:    Sample $\{\boldsymbol{w}_h\}_{h=1}^M$, i.i.d. from $\pi$.
7:    $\forall h \in [M]$ and $j \in [m]$, calculate $\mathbf{L}_{\boldsymbol{w}_h}(\theta_j^*(t)) = \boldsymbol{w}_h^T \mathbf{L}(\theta_j^*(t))$.
8:    $\forall h \in [M]$, calculate $u_{\boldsymbol{w}_h}(t)$ defined in (12) for each $h$.
9:  **else**
10:    $u_{\boldsymbol{w}_h}(t) = u_{\boldsymbol{w}_h}(t-1)$
11:  **end if**
12:  $\forall j \in [m]$, update $\theta_j^*(t)$ by (13). (May use mini-batch gradient defined in (16)).
13: **end for**

---

**Methods to construct confidence interval** The methods we used to construct confidence interval are – The Normal interval:

$$[\hat{\theta} - z((1+\alpha)/2)\hat{\text{se}}_{\text{boot}}, \hat{\theta} + z((1+\alpha)/2)\hat{\text{se}}_{\text{boot}}],$$

where $z(\cdot)$ is the inverse cumulative distribution function of standard Normal distribution. And $\hat{\text{se}}_{\text{boot}}$ is the standard deviation estimated from $\tilde{\rho}_\pi$.

– The percentile intervals:

$$[Q[(1-\alpha)/2, \tilde{\rho}_\pi], Q[(1+\alpha)/2, \tilde{\rho}_\pi]].$$

– The pivotal interval:

$$[2\hat{\theta} - Q[(1+\alpha)/2, \tilde{\rho}_\pi], 2\hat{\theta} - Q[(1-\alpha)/2, \tilde{\rho}_\pi]].$$

We consider the following simple linear regression: $x \sim \mathcal{N}(\boldsymbol{0}, \boldsymbol{I})$, $y \mid x \sim \mathcal{N}(\theta^\top x, \boldsymbol{I})$, where the features $x \in \mathbb{R}^4$ and we set the true parameter to be $\theta_0 = [1, -1, 1, -1]$. We consider $n = 50$ and the number of particles $m = 20, 50, 100, 200$. We compare the coverage probability and the confidence level $\alpha$ to measure the quality:

**Measuring the quality of confidence interval** With a large number $N$ of independently generated training data (we use $N = 1000$), we are able to obtain the corresponding confidence intervals $\{\text{CI}(\alpha)_s\}_{s=1}^N$ and thus obtain the probability that the true parameter falls into the confidence intervals, which is the estimated coverage probability

$$\hat{\alpha} = \frac{1}{N} \sum_{s=1}^N \mathbb{I}\{\theta_0 \in \text{CI}(\alpha)_s\}.$$

A good confidence interval should have $\hat{\alpha}$ close to $\alpha$. Thus we measure the performance by calculating the difference $|\alpha - \hat{\alpha}|$.

As $\hat{\theta}_{\boldsymbol{w}}$ is the least square estimator of the bootstrapped dataset, it has analytic solution and thus can be obtained via some matrix multiplications. $\theta_{\boldsymbol{w}}^*$ is initialized using $\hat{\theta}_{\boldsymbol{w}}$ and then updated for 2000 steps. For this experiment, we find that adding the threshold $\gamma$ does not gives further improvement for this experiment and thus we simply set $\gamma = 0$ and use $M = 1$. We approximate the true bootstrap distribution by sampling 10000 i.i.d. samples.

**More experiment result** Table 3 all the result we have varying $\alpha = 0.8, 0.9, 0.95$, $m = 20, 50, 100, 200$ and three different approaches for constructing confidence interval. As we can see, centroid approximation gives the best performance in most cases compared with the other three baselines.

| | Num Particle | | 20 | 50 | 100 | 200 |
|---|---|---|---|---|---|---|
| $\alpha = 0.8$ | Normal | Bootstrap | $0.033 \pm 0.013$ | $0.028 \pm 0.013$ | $0.026 \pm 0.013$ | $0.031 \pm 0.013$ |
| | | Bayesian | $0.084 \pm 0.014$ | $0.076 \pm 0.014$ | $0.082 \pm 0..014$ | $0.086 \pm 0.014$ |
| | | Residual | $\mathbf{0.033 \pm 0.013}$ | $0.037 \pm 0.013$ | $0.029 \pm 0.013$ | $\mathbf{0.024 \pm 0.013}$ |
| | | Centroid | $0.036 \pm 0.013$ | $\mathbf{0.003 \pm 0.012}$ | $\mathbf{0.017 \pm 0.013}$ | $0.030 \pm 0.013$ |
| | Percentile | Bootstrap | $0.096 \pm 0.014$ | $0.050 \pm 0.014$ | $0.044 \pm 0.013$ | $0.024 \pm 0.013$ |
| | | Bayesian | $0.114 \pm 0.015$ | $0.079 \pm 0.014$ | $0.074 \pm 0.014$ | $0.071 \pm 0.014$ |
| | | Residual | $0.079 \pm 0.014$ | $0.032 \pm 0.013$ | $\mathbf{0.017 \pm 0.013}$ | $\mathbf{0.010 \pm 0.013}$ |
| | | Centroid | $\mathbf{0.066 \pm 0.014}$ | $\mathbf{0.008 \pm 0.013}$ | $0.019 \pm 0.013$ | $0.020 \pm 0.013$ |
| | Pivotal | Bootstrap | $0.101 \pm 0.015$ | $0.053 \pm 0.014$ | $0.045 \pm 0.014$ | $0.033 \pm 0.013$ |
| | | Bayesian | $0.158 \pm 0.015$ | $0.110 \pm 0.110$ | $0.088 \pm 0.014$ | $0.078 \pm 0.014$ |
| | | Residual | $0.087 \pm 0.014$ | $0.044 \pm 0.013$ | $0.030 \pm 0.013$ | $\mathbf{0.023 \pm 0.013}$ |
| | | Centroid | $\mathbf{0.026 \pm 0.013}$ | $\mathbf{0.030 \pm 0.012}$ | $\mathbf{0.018 \pm 0.013}$ | $0.030 \pm 0.013$ |
| $\alpha = 0.9$ | Normal | Bootstrap | $0.029 \pm 0.010$ | $0.031 \pm 0.011$ | $0.021 \pm 0.010$ | $0.017 \pm 0.010$ |
| | | Bayesian | $0.076 \pm 0.012$ | $0.054 \pm 0.011$ | $0.048 \pm 0.011$ | $0.045 \pm 0.011$ |
| | | Residual | $0.043 \pm 0.011$ | $0.023 \pm 0.010$ | $0.025 \pm 0.010$ | $0.020 \pm 0.010$ |
| | | Centroid | $\mathbf{0.027 \pm 0.010}$ | $\mathbf{0.001 \pm 0.009}$ | $\mathbf{0.012 \pm 0.010}$ | $\mathbf{0.016 \pm 0.010}$ |
| | Percentile | Bootstrap | $0.101 \pm 0.013$ | $0.036 \pm 0.011$ | $0.021 \pm 0.010$ | $\mathbf{0.014 \pm 0.010}$ |
| | | Bayesian | $0.129 \pm 0.013$ | $0.077 \pm 0.012$ | $0.059 \pm 0.012$ | $0.054 \pm 0.011$ |
| | | Residual | $0.098 \pm 0.013$ | $0.030 \pm 0.011$ | $0.033 \pm 0.011$ | $0.025 \pm 0.010$ |
| | | Centroid | $\mathbf{0.081 \pm 0.012}$ | $\mathbf{0.021 \pm 0.010}$ | $\mathbf{0.020 \pm 0.010}$ | $0.015 \pm 0.010$ |
| | Pivotal | Bootstrap | $0.106 \pm 0.013$ | $0.045 \pm 0.011$ | $0.025 \pm 0.010$ | $0.023 \pm 0.010$ |
| | | Bayesian | $0.149 \pm 0.014$ | $0.093 \pm 0.012$ | $0.073 \pm 0.012$ | $0.056 \pm 0.011$ |
| | | Residual | $0.100 \pm 0.013$ | $0.044 \pm 0.011$ | $0.030 \pm 0.011$ | $0.023 \pm 0.010$ |
| | | Centroid | $\mathbf{0.046 \pm 0.011}$ | $\mathbf{0.013 \pm 0.009}$ | $\mathbf{0.011 \pm 0.010}$ | $\mathbf{0.020 \pm 0.010}$ |
| $\alpha = 0.95$ | Normal | Bootstrap | $\mathbf{0.018 \pm 0.008}$ | $0.014 \pm 0.008$ | $0.012 \pm 0.008$ | $0.006 \pm 0.007$ |
| | | Bayesian | $0.053 \pm 0.010$ | $0.038 \pm 0.009$ | $0.031 \pm 0.009$ | $0.037 \pm 0.009$ |
| | | Residual | $0.036 \pm 0.009$ | $0.019 \pm 0.008$ | $0.011 \pm 0.008$ | $0.008 \pm 0.007$ |
| | | Centroid | $\mathbf{0.018 \pm 0.008}$ | $\mathbf{0.005 \pm 0.006}$ | $\mathbf{0.009 \pm 0.007}$ | $\mathbf{0.005 \pm 0.007}$ |
| | Percentile | Bootstrap | $0.081 \pm 0.010$ | $0.047 \pm 0.009$ | $0.030 \pm 0.008$ | $0.017 \pm 0.008$ |
| | | Bayesian | $0.126 \pm 0.012$ | $0.072 \pm 0.010$ | $0.056 \pm 0.010$ | $0.042 \pm 0.009$ |
| | | Residual | $0.100 \pm 0.011$ | $0.040 \pm 0.009$ | $0.037 \pm 0.009$ | $0.021 \pm 0.008$ |
| | | Centroid | $\mathbf{0.077 \pm 0.010}$ | $\mathbf{0.029 \pm 0.008}$ | $\mathbf{0.020 \pm 0.008}$ | $\mathbf{0.016 \pm 0.008}$ |
| | Pivotal | Bootstrap | $0.089 \pm 0.011$ | $0.043 \pm 0.009$ | $0.027 \pm 0.008$ | $0.015 \pm 0.008$ |
| | | Bayesian | $0.127 \pm 0.012$ | $0.091 \pm 0.011$ | $0.064 \pm 0.010$ | $0.056 \pm 0.010$ |
| | | Residual | $0.085 \pm 0.011$ | $0.051 \pm 0.009$ | $0.036 \pm 0.009$ | $0.029 \pm 0.008$ |
| | | Centroid | $\mathbf{0.046 \pm 0.009}$ | $\mathbf{0.002 \pm 0.007}$ | $\mathbf{0.014 \pm 0.008}$ | $\mathbf{0.009 \pm 0.007}$ |

Table 3: Complete result on comparing centroid approximation with various bootstrap methods. The bold number shows the best approach.

---

**Algorithm 3** Algorithm for Centroid Approximation Applied to Contextual Bandit.

1: Obtain a randomly initialized $\theta_j^*(0)$, $j \in [m]$.
2: Initialize a common replay buffer $R_c = \emptyset$ recording all the observed contexts.
3: For each model, initialize its own replay buffer $R_j = \emptyset$ that is used for training.
4: **for** $t \in$ number of total steps **do**
5:      Obtain the $t$-th context $x_t$.
6:      Sampling one model based on probability $\{v_j^*(t)\}_{j=1}^m$ to make action $a_t$ and get reward $r_t$.
7:      Update the common replay buffer by $R_c \leftarrow R_c \cup \{(x_t, a_t, r_t)\}$
8:      // Update $\boldsymbol{w}_h$ and $R_j$ and model every a few iterations.
9:      **if** $t$ mod freq $== 0$ **then**
10:          $\forall j \in [m]$, calculate $\mathbf{L}(\theta_j^*(t))$ defined in (11) for all the contexts in $R_c$. // $\mathbf{L}(\theta_j^*(t)) \in \mathbb{R}^{|R_c|}$.
11:          Generate $M$ sets of random weights $\{\boldsymbol{w}_h\}_{h=1}^M$ of contexts in $R_c$ from $\pi$.
12:          $\forall h \in [M]$ and $j \in [m]$, calculate $\mathbf{L}_{\boldsymbol{w}_h}(\theta_j^*(t)) = \boldsymbol{w}_h^T \mathbf{L}(\theta_j^*(t))$.
13:          $\forall h \in [M]$, calculate $u_{\boldsymbol{w}_h}(t)$ defined in (12) for each $h$.
14:          $\forall i \in [|R_c|]$ and $j \in [m]$, calculate $q_{i,j}$ by (15)
15:          $\forall j \in [m]$, update $v_j^*(t)$ based on (7).
16:          $\forall j \in [m]$, if $v_j^*(t) \leq \gamma$, construct $R_j = R_c$, else, construct $R_j$ by sample $|R_c|$ contexts
       in $R_c$. The probability that context $i$ is being sampled is $q_{i,j} / \sum_{i=1}^{|R_c|} q_{i,j}$.
17:          $\forall j \in [m]$, train model $j$ using the data in $R_j$ for several iterations.
18:      **end if**
19: **end for**

---

## C.2 CENTROID APPROXIMATION FOR BOOTSTRAP METHOD IN CONTEXTUAL BANDIT

### C.2.1 MORE BACKGROUND

Contextual bandit is a classic task in sequential decision making problem in which at time $t = 1, ..., n$, a new context $x_t$ arrives and is observed by an algorithm. Based on its internal model, the algorithm selects an actions $a_t$ and receives a reward $r_t(x_t, a_t)$ related to the context and action. During this process, the algorithm may update its internal model with the newly received data. At the end of this process, the cumulative reward of the algorithm is calculated by $r = \sum_{t=1}^n r_t$ and the goal for the algorithm is to improve the cumulative reward $r$. The exploration-exploitation dilemma is a fundamental aspect in sequential decision making problem such as contextual bandit: the algorithm needs to trade-off between the best expected action returned by the internal model at the moment (i.e., exploitation) with potentially sub-optimal exploratory actions. Thompson sampling (Thompson, 1933; Wyatt, 1998; May et al., 2012) is an elegant and effective approach to tackle the exploration-exploitation dilemma using the model uncertainty, which can be approached with various methods including Bayesian posterior (Graves, 2011; Welling & Teh, 2011), dropout uncertainty (Srivastava et al., 2014; Hron et al., 2017) and Bootstrap (Osband et al., 2016; Hao et al., 2019). The ability to accurately assess the uncertainty is a key to improve the cumulative reward. Bootstrap method for contextual bandit maintains $m$ bootstrap models trained with different bootstrapped training set. When conducting an action, the algorithm uniformly samples a model and then selects the best action returned by the sampled model.

### C.2.2 MORE EXPERIMENT SETUP DETAILS

We set all the experimental setting including data preprocessing, network architecture and training pipeline exactly the same as the one used in Riquelme et al. (2018) and adopt its open source code repository.

**Network architecture** Following Riquelme et al. (2018), we consider a fully connected feed forward network with two hidden layers with 50 hidden units and ReLU activations. The input and output dimensions depends on the dimension of context and number of possible actions.

**Training** For each dataset, we randomly generate 2000 contexts, and for each algorithm, we update the replay memory buffer for each model every 50 contexts, and each model is also updated every 50 contexts. For the standard bootstrap, when updating the replay buffer of each model, we sample 50 i.i.d. contexts with uniform probability from the latest 50 contexts (each model have different

realizations of the samples) and add the newly sampled contexts to each model's replay buffer. For the centroid approximation, we update the replay buffer of each model by applying resampling on all the observed contexts up to the current steps. The resampling probability of each context for each model is different and determined by the algorithm. We refer readers to Algorithm 3 for the detailed procedures. Here we choose freq $= 50$ and $M = 100$. When at model updating, each model is trained for 100 iterations with batch size 512 using the data from its replay buffer. Following Riquelme et al. (2018), we use RMSprop optimizer with learning rate 0.1 for optimizing. When making actions, we sample the prediction head according to $v_k^*(t)$ obtained using the examples between the last two model updates.

Notice that in the implementation, we only need to maintain one common replay buffer and the replay buffer for each model can be implemented by maintaining the number of each context. Thus when sampling batches of context, we simply need to sample the index of the context and refer to the common replay buffer to get the actual data.

## C.3 Centroid Approximation for Bootstrap DQN

### C.3.1 More Background

Similar to the bootstrap method in contextual bandit problem, Bootstrap DQN explores using the model uncertainty, which can be assessed via maintaining several models trained with bootstrapped training set. Maintaining several independent models can be very expensive in RL and to reduce the computational cost, Bootstrap DQN uses a multi-head network with a shared base. Each head in the network corresponds to a bootstrap model and the common shared base is thus trained via the union of the bootstrap training set of each head. We train the Bootstrap DQN with standard updating rule for DQN and use Double-DQN (Van Hasselt et al., 2016) to reduce the overestimate issue. Notice that our centroid approximation method only changes the memory buffer for each head and thus introduces no conflict to other possible techniques that can be applied to Bootstrap DQN.

### C.3.2 More experiment setup details

**Network Architecture** Following Osband et al. (2016), we considered multi-head network structure with a shared base layer to save the memory. Specifically, we use a fully connected layer with 256 hidden neurons as the shared base and stack two fully connected layers each with 256 hidden neurons as head. Each head in the model can be viewed as one bootstrap particles and in computation, all the bootstrap particles use the same base layer.

**Training and Evaluation** For LunarLander-v2, we train the model for 450 episodes with the first 50 episodes used to initialize the common memory buffer. The maximum number of steps within each episode is set to 1000 and we report the moving average reward with window width 100. For Catcher-v0, we train the model for 100 episodes with the first 10 episodes used to initialize the common memory buffer. We set the maximum number of steps within each episodes 2000 and report the moving average reward with window width 25.

For training the Bootstrap DQN, given the current state $x_t$, we sample one particle based on $\{v_j^*\}_{j=1}^m$ and use its policy network to make an action $a_t$ and get the reward $r_t$ and next state $x_{t+1}$. The Q-value of the state action pair $Q(x_t, a_t)$ is estimated by $r_t + \lambda * \hat{Q}(x_{t+1})$, where $\hat{Q}(x_{t+1})$ is the predicted state value by the target network of the sampled particle and $\lambda$ is the discount factor set to be 0.99. At each step, the policy network of all particles are updated using one step gradient descent with Adam optimizer ($\beta = (0.9, 0.999)$ and learning rate 0.001) and mini-batch data (size 64) sampled from its replay buffer. We update target model, each particle's replay buffer and $v_j^*$s every 1000 steps for LunarLander-v2 and every 200 steps for Catcher-v0. The update scheme for replay buffers of each particles and $v_j^*$s is the same as the one in contextual bandit experiment. As the model see significantly larger number of contexts than that in the contextual bandit experiment, to reduce the memory consumption, we set the max capacity of the common replay buffer to 50000 (the oldest data point will be pop out when the size reaches maximum and new data comes in). For training the shared base, following Osband et al. (2016), we adds up all the gradient comes from each head and normalizes it by the number of heads. Algorithm 4 summarizes the whole training pipeline.

---

**Algorithm 4** Algorithm for Centroid Approximation Applied to DQN.

---

1: Obtain a randomly initialized $\theta_j^*(0)$, $j \in [m]$. (For the $j$-th particle, both of its target and policy network use the same initialization.)
2: Initialize a common replay buffer $R_c = \emptyset$ recording all the observed contexts.
3: For each head, initialize its own replay buffer $R_j = \emptyset$ that is used for training.
4: **for** $t \in$ number of total episodes **do**
5:      **while** not at terminal state and the number of steps does not exceed the threshold **do**
6:          Obtain the $t$-th context $x_t$.
7:          Sample an head based on $\{v_j^*\}$ to make action $a_t$ and get $Q(x_t, a_t)$ using the reward $r_t$ and the prediction of the corresponding target network.
8:          Update the common replay buffer by $R_c \leftarrow R_c \cup \{(x_t, Q(x_t, a_t))\}$
9:          $\forall j \in [m]$, update its policy network by one step gradient descent using the data from its reply buffer.
10:          // Update $\boldsymbol{w}_h$ and $R_j$ and target network every a few iterations.
11:          **if** $t$ mod freq $== 0$ **then**
12:             $\forall j \in [m]$, calculate $\mathbf{L}(\theta_j^*(t))$ defined in (11) for all the contexts in $R_c$.
13:             Generate $M$ sets of random weights $\{\boldsymbol{w}_h\}_{h=1}^M$ of contexts in $R_c$ from $\pi$.
14:             $\forall h \in [M]$ and $j \in [m]$, calculate $\mathbf{L}_{\boldsymbol{w}_h}(\theta_j^*(t)) = \boldsymbol{w}_h^T \mathbf{L}(\theta_j^*(t))$.
15:             $\forall h \in [M]$, calculate $u_{\boldsymbol{w}_h}(t)$ defined in (12) for each $h$.
16:             $\forall i \in [|R_c|]$ and $j \in [m]$, calculate $q_{i,j}$ by (15)
17:             $\forall j \in [m]$, update $v_j^*(t)$ based on (7).
18:             $\forall j \in [m]$, if $v_j^*(t) \le \gamma$, construct $R_j = R_c$, else, construct $R_j$ by sample $|R_c|$ contexts in $R_c$. The probability that context $i$ is being sampled is $q_{i,j}/\sum_{i=1}^{|R_c|} q_{i,j}$.
19:             $\forall j \in [m]$, update the $j$-th target network by loading the weights of the $j$-th policy network.
20:          **end if**
21:      **end while**
22: **end for**

---

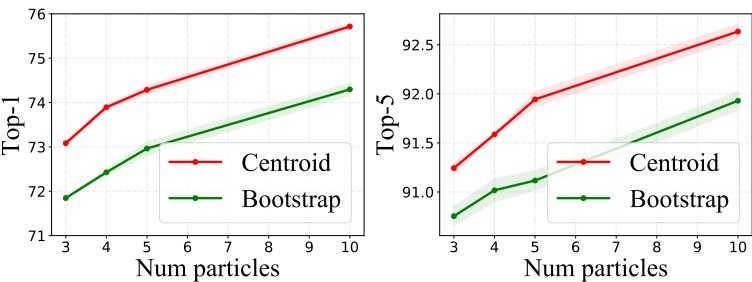

Figure 4: Results on ensemble modeling with bootstrap using Vgg16 on CIFAR-100.

## C.4 BOOTSTRAP ENSEMBLE MODEL

Ensemble of deep neural networks have been successfully used to boost predictive performance (Lakshminarayanan et al., 2017). In this experiment, we consider using an ensemble of deep neural network trained on different bootstrapped training set, which is also known as a popular strategy called *bagging*.

We consider image classification task on CIFAR-100 and use standard VGG-16 (Simonyan & Zisserman, 2014) with batch normalization. We apply a standard training pipeline. We train the bootstrap model for 160 epochs using SGD optimizer with 0.9 momentum and batchsize 128. The learning rate is initialized to be 0.1 and is decayed by a factor of 10 at epoch 80 and 120. We start to apply the centroid approximation at epoch 120 (thus the centroid is initialized with 120 epochs' training). We generate the bootstrap training set for each centroid every epoch using the proposed centroid approximation method. We consider $m = 3, 4, 5, 10$ ensembles and use $\gamma = 0.5/m$. We

| #Particle | $\gamma = 0$ | $\gamma = 0.5$ | $\gamma = m$ |
|---|---|---|---|
| 3 | $3480.0 \pm 120$ | $3702.7 \pm 89.8$ | $3467.7 \pm 115$ |
| 4 | $3461.92 \pm 126$ | $3723.1 \pm 78.7$ | $3600.0 \pm 69.3$ |
| 5 | $3586.5 \pm 64.5$ | $3799.6 \pm 84.2$ | $3647.3 \pm 64.5$ |
| 10 | $3785.0 \pm 59.1$ | $3796.9 \pm 36.1$ | $3742.7 \pm 86.8$ |

Table 4: Ablation study.

repeat the experiment for 3 random trials and report the averaged top1 and top5 accuracy with the standard deviation of the mean estimator. Algorithm

Figure 4 summarizes the result. Overall, increasing $m$ is able to improve the predictive performance and with the same number of models, our centroid approximation consistently improves over standard bootstrap ensembles.

## C.5 ABLATION STUDY

We study the effectiveness of using (8) to modify the gradient of centroid with $v_k^*(t) \leq \gamma$. We consider the setting $\gamma = 0$ (no modification) and $\gamma = m$ (always modify, equivalent to no bootstrap uncertainty) and applied the method on the mushroom dataset in the contextual bandit problem. Table 4 shows that (i) modifying the gradient of centroid with small $v_k^*(t)$ using do improve the overall result; (2) bootstrap uncertainty is important for exploration.

## C.6 COMPUTATION OVERHEAD

Our main goal is not to decrease the training cost but improve the quality of bootstrap partical distribution so that we can use less models at deployment and hence reduce the memory cost and the computational cost for inference. Actually, as discussed in Section 3, our method actually only introduces a little computation overhead while much improves the quality of the particles, which is another advantage of our method. For example, in bandit problem on mushroom dataset, when $m = 3, 10$, vanilla bootstrap takes $33s, 101s$ while our approach takes $35s, 108s$ per run. For the bagging, when $m = 3, 10$, vanilla bootstrap takes $11200s, 34240s$ while ours take $12000s, 36200s$. Results are based on the average of 3 runs. Our method only introduces about $7\%$ computational overhead even with an naive implementation.

