# OpenReview forum: "Centroid Approximation for Bootstrap"
_ICLR.cc/2022/Conference — ICLR 2022 Submitted_

### Official Review · Reviewer_Uq9L · 2021-10-20

**Correctness:** 3
**Technical Novelty And Significance:** 3
**Empirical Novelty And Significance:** 2
**Recommendation:** 5
**Confidence:** 4

**Main Review:**

##########################################################################

Summary:
This paper offers an interesting approach to scaling bootstrapping which is applicable to many machine learning problems. While it is not a perfect method, I think a paper providing a thorough evaluation of this approach and the previous ones is worthy of publication, if done properly.


##########################################################################

Reasons for score:
For the most part, the paper is written clearly but some of the crucial comparisons against previous approaches (see below) are missing. I am slightly leaning toward reject since the paper probably needs more work. While the algorithm is novel, it is unclear how much benefit it would offer above previous approaches. I am open to adjusting this score in the rebuttal stage, pending the authors' responses.


##########################################################################

Pros:


1. The paper addresses an interesting computational problem applicable to many models in machine learning. The algorithm is simple to implement and to understand.

2. Experiments cover interesting machine learning use cases, e.g. contextual bandit, reinforcement learning, confidence interval estimation.


##########################################################################

Cons:


1. It is unclear how much benefit the new methodology offers over the previous ones, e.g. in terms of model accuracy improvement.
It is presumed that the iterative nature of the algorithm incurs many model updates and model loss evaluations in the inner loop.

2. Comparison against previous approaches except for vanilla bootstrap are largely missing. It would be beneficial to comment on how this method compares against m-out-of-n bootstrap and bag-of-little bootstraps. The authors mention that both of these methods are inherently slower (page 9, Section 6). However, these methods are highly parallelizable algorithms and they are worthy of discussion (unlike the iterative algorithm proposed in this paper).

3. As stated in the appendix, the motivation of the paper is not to decrease the training cost but reduce the memory cost and the computational cost for inference. How much compression can be achieved in terms of memory? It seems the study on establishing the relationship between optimal m and n is for future work but it is a bit unclear from the experiments at least empirically.

4. How big is the dataset (n) in the experiments? Any comment on how the proposed method performs for ensemble tree methods, as the experiments discuss primarily neural networks? Please also comment on the dimensionality of theta in each experiment setting.

5. On a related above: how would the initialization of theta work for a tree based method?

##########################################################################

Questions during rebuttal period:


Please provide clarifications for the points in the previous section.



#########################################################################

Some typos:

- page 19: bootstrap partical -> bootstrap particle


**Summary Of The Paper:**

This paper proposes an acceleration technique for bootstrap. Here a bootstrap particle corresponds to a machine learning model and for a large number of bootstrap particles, a huge computational cost is incurred. The authors propose using a k-means-like centroid approximation which optimizes a set of centroids (each of which corresponds to a machine learning model) minimizing a data-dependent Wasserstein distance.

**Summary Of The Review:**

It is overall an interesting paper providing an acceleration technique for bootstrap - but I would suggest the following:
1. Provide a clarification in the beginning of this paper, rather than in the appendix, that this paper focuses on "model compression", hence inference time improvement and model size management, rather than reducing the training time.

2. Clarify experimental setting parameters (e.g. size of the dataset, size of theta, etc.). See above for details.

3. Comment on comparison vs other previous bootstrap variants.

---

> ### Author Response · Authors · 2021-11-22
> **Thanks reviewer Uq9L for the comments! Below please find our response.**
>
> **Regarding model accuracy improvement**
> We are not very sure whether we understand your comments on "model accuracy". The main focus is to improve the uncertainty quantification when the number of bootstrap particles is limited and our experiments are designed to demonstrate this improvement. In terms of the improvement of the prediction accuracy, the bagging experiment in appendix C4 might be the one you are looking for?
>
> **Regarding comparing with Bootstrap variants**
> We think our paper and m-out-of-n bootstrap/bag-of-little bootstraps are working towards two orthogonal directions of improving the scalability of bootstrap and thus not comparable. Our paper aims to improve the accuracy of bootstrap when a limited number of bootstrap particles are allowed at *inference*. While m-out-of-n bootstrap/bag-of-little bootstraps aim to decrease the *training* cost when the size of the dataset is large. On the other hand, we should not expect m-out-of-n bootstrap/bag-of-little bootstraps is able to give better performance at inference time, and hence comparing with the standard bootstrap should be viewed as a stronger baseline than its approximated version.
>
>
> **Regarding memory compression**
> There are two ways to evaluate the model compression. 1. Under same ‘accuracy’, how many particles are needed; 2. Using the same number of particles, how much ‘accuracy’ is improved. We mainly use the second way as the change of particles is discrete and hence not that convenient to quantify. Note that we can still get the information of the 1st evaluation using our experiment result. For example in Table 2 in the mushroom dataset, m=3 of our centroid approach has better performance than m=10 vanilla bootstrap showing a >= 3.3x model compression. Similarly, in Table 1 m=50 centroid approach is better than the m=200 vanilla bootstrap suggesting >= 4x model compression. Such comparison can be inferred similarly in another result.
>
> Thanks for the suggestion to mention we are improving the inference time cost at the very beginning of the paper. Actually, we’ve emphasized it at the very beginning of the paper (footnote 2). We add an extra sentence (in the third paragraph of the introduction) to highlight this point in the updated manuscript.
>
> **Regarding the size of the dataset**
> For the contextual bandit, we consider 2000 contexts (see Appendix C2.2). The DQN experiment is a reinforcement learning setting and the total number of episodes is introduced in Appendix C3.2. The bagging experiment uses cifar-100 with 50000 training data points and 10000 testing data points.
>
> **Regarding compared with ensemble tree method**
> Our approach is model agnostic and hence can be applied to tree-based approach. We can use both random initializations or initialize with a random bootstrap particle for a tree-based approach.
>
> **Regarding the dimensionality for neural network**
> For the bandit experiment, there are about 3000 trainable parameters. For the DQN, it's about 70K parameters. For bagging, vgg-16 has about 140M parameters.

---

### Official Review · Reviewer_oMBb · 2021-10-29

**Correctness:** 3
**Technical Novelty And Significance:** 3
**Empirical Novelty And Significance:** 3
**Recommendation:** 5
**Confidence:** 4

**Main Review:**

Strengths:

1) The problem is important and well-stated. The paper is clearly written.
2) The arguments justifying the algorithmic steps and the theory of Section~4 are plausible for the most commonly observed scenario in data sciences when hyperparameter dimension is fixed (and low).

Weakness:

1) I am surprised that the hyperparameter dimension plays no roll in either the assumptions or the results of Section4. This is not the case for several of the cited references, and requires some clarification. For example, when hyperparameter dimension grows, Assumption~2 may not be valid without proper control over $d$, or additional assumptions (like sparsity of $\theta$). If the authors are working in the fixed-$d$ case, that should be clarified.

2) The algorithm lacks clarity. In particular, I am confused about exactly how the \textit{spread} of $\rho_{\pi}$ and its tails and extremes are being captured. Considerable additional discussions and clarifications are needed about this.

3) How is $m$ chosen? Is this optimized in any way? It seems that Algorithm~1 of page 16 can be very sensitive to the choice of $m$. The algorithm also requires (at least) $M$ iid draws from $\pi$, so what is $M$ and how is this chosen?


4) The algorithm/theory seems to require that the randomly-weighted log-likelihood $L_w$ should have sufficient smoothness properties near each centroid. Since centroids can be anywhere in the support of the hyperparameter, this requires strong conditions on $L_w$. One important property that seems to be needed is a locally quadratic behavior, ie, equation (5) needs to hold at every centroid. This seems to be a very strong condition. This is different from the traditional bootstrap conditions, where similar properties are needed only near the true value of the hyperparameter.


5) Bootstrap is computationally expensive but also embarrassingly parallel. So, while still challenging in applications where there is limited computational resource, the computational burden of bootstrap may be expected to  become less of an issue with time.

6) 2 lines after equation (4): "We emphasize that the optimal solution to two-stage learning is guaranteed to be the global minimizer of the loss in (2)." Where are you getting this from?

7) I would guess that different centroids "stabilize" at different rates, and some centroids are much easier to locate than others. The algorithm seems to treat all centroids equally. Some discussion would be helpful about this.


**Summary Of The Paper:**

Bootstrap is a versatile technique for uncertainty quantification and a viable alternative to Bayesian inference, but it can be expensive in terms of memory and computations. With a small number of particles/samples, bootstrap may perform poorly. As a consequence, bootstrap can be quite expensive and even unaffordable for deep learning problems with huge models. The paper proposes to address the question of improving the accuracy of bootstrap inference when the number of particles is limited.

The approach is to minimize the Wasserstein distance between the ideal bootstrap distribution (denoted by $\rho_{\pi}$ in the paper) and a probability mass mass function supported on a small number of particles (called "centroids" in the paper). The exposition given around equations (3) and (4) plausible decomposes this problem into a coupled two-stage optimization problem. Unfortunately, $\rho_{\pi}$ is not observe, which is a major challenge to this conceptual development. The paper then relies on the fact that in standard problems when training data size is large, the bootstrap obtains properties similar (or identical) to the MLE or Bayes estimators, and the algorithmic steps are base don this framework.


**Summary Of The Review:**

The bootstrap is an extremely versatile technique, and is applicable in many situations, for example, with dependent data, on extremes, cases where standard frequentist or Bayesian asymptotics may not hold, and so on. While the present papers approach does not cover many of these bases, it does address the most commonly observed scenario: where standard mathematical/calculus/probabilistic properties hold (like interchanges of limits and integrals etc) and the hyperparameter estimator has a limiting Gaussian distribution due to the Central Limit Theorem. This is the most important special case to tackle, and as an early attempt to address the computational aspects of bootstrap, I think this paper is showing quite a bit of promise.


Despite the long list of weaknesses mentioned above,  my initial feeling about this paper is quite positive. Bootstrap is a very strong alternative to Bayesian or ad hoc methods used in ML, and this paper may be a breakthrough to address the computational challenges of bootstrap. I am excited about this, but would appreciate considerable additional details and clarifications.


Revised comments: I thank the authors for the updates on the paper and their detailed responses. While I am a strong enthusiast of the bootstrap and I like this paper, I still have several concerns. It would take a major revision to address these concerns, so I am leaving my rating unchanged.

1) The bootstrap essentially is a technique for uncertaintly quantification, so it's main computational burden is during training: I am not convinced about the realism or practicality of the authors' premise that there is unlimited training resouce/time but limited time during inference. In any case, during inference, the bootstrap is just a bunch of forward model runs: for example in deep learning, it would be just evaluating what we get from the architecture for a given set of weights and biases. It certainly helps if the "set of weights and biases" is small, but I am struggling to find a realistic example where it is hard to simply run the code for a given architecutre with given weights and biases and test set features.

2) The roles of m and M need to be studied more carefully.

3) I am not convinced that the centroid method sufficiently addresses the issue of variability or stochasticity in the hyperparameter space. We need proper studies that the tails of the distribution are adequtely captured using the centroids method, which will take more work than what the paper currently contains.

---

> ### Author Response · Authors · 2021-11-22
> **Thanks reviewer oMBb for the comments! Below please find our response**
>
> **Regarding the parameter dimension**
> Yes, we don’t work on the high dimensional setting and assume $d$ is fixed and not scale with $n$. We update the manuscript to clarify this (see the last sentence of the 2nd paragraph in section 4).
>
> **Regarding the clarity of the algorithm**
> Our algorithm is $\rho_\pi$-agnostic but in theory, we require asymptotic in order to establish the connection with Wasserstein distance minimization. We also update the ‘Practical algorithm’ and Appendix B to improve the clarity of the algorithm.
>
> **Regarding how to choose m and M**
> m is the number of bootstrap particles we need to use to approximate the ideal bootstrap distribution. The larger m means better approximation (this is like adding more samples can improve the quality of the empirical distribution) but more resource consumption. In practice, the actual choice of m is problem and resource-dependent so our goal is to show the improvement for different choices of m.
> M is the number of Monte Carlo samples of $w$ we need to have in order to approximate $g$ in (8). As discussed in the paper, we choose M=100 for all the experiments. In practice, the choice of M is not sensitive at all once it’s large enough. Also, as shown in ‘Practical algorithm’, increasing M does not introduce much training computation overhead.
>
> **Regarding equ (5)**
> Equ (5) actually holds with high probability when $n$ is large, which is characterized by Theorem 1. Note that the assumption for theorem 1 is almost the same assumption for the standard bootstrap theory (except that we assume a high-level assumption 2, which holds under some weak regularity conditions for bootstrap).
>
> **Regarding the training cost**
> Yes, we agree that bootstrap can be better parallelized than our approach. But as mentioned in footnote 2, the training cost can be negligible than inference cost which is the main focus of this paper.
>
> **Regarding 2 lines after Equ(4)**
> Thanks for the comments! It’s can be concluded directly from Lemma 3.1 and 3.2 in [1]. We also update the manuscript and give a pointer to whether this result is from.
>
> **Regard different rates of stabilizing**
> Thanks for the comments. We think the different stabilize rate is actually one case that can cause the centroid degeneration phenomenon, in which one centroid might be harder to locate, and hence its performance is always worse than others. Our technique to prevent centroid degeneration is able to well handle this situation.
>
> [1] Canas, Guillermo D., and Lorenzo Rosasco. "Learning probability measures with respect to optimal transport metrics.

---

### Official Review · Reviewer_5nLX · 2021-11-02

**Correctness:** 3
**Technical Novelty And Significance:** 1
**Empirical Novelty And Significance:** 2
**Recommendation:** 5
**Confidence:** 3

**Main Review:**

The major points can be summarised as follows:
1. While the concept and idea is nice, I think they are not entirely novel. k-means has often been used as a trick to summarise distributions with smaller samples. There are other methods such as Coresets (https://arxiv.org/abs/1906.03329) which also provide distributional approximations with smaller samples. How does this method compare to these prevailing methods?  This can be discussed in detail with either experimental or theoretical justifications. While the authors do a good job in terms of depicting the idea, these comparisons I find are a bit lacking. Suppose the data is derived from a mixture model. In this case, does this method capture the relevant modes (assuming the number of components is known). Do the obtained centroids provide a consistent estimation of the mixture component parameters? Can you provide an experiment to verify that? How much do the results depend on the starting seeds?

 2. Additionally, I cannot quite understand some of the intuition in the paper. For example, do the results (Theorem 2) hold for any distribution $\pi$ on $w$ or is this specific to the multinomial distribution? Does the approximation in Eq. 5 hold for any $w$. It is unclear to me what the intuition is behind such a logic? I do not understand why $\mathscr{L}_{\infty} \approx  \mathscr{L}_w$, or why the approximation for the corresponding estimators of $\theta$ hold. I think the paper tries to argue that different centroids of the Bootstrap samples capture different regions of the distribution, but I do not completely follow the theoretical justification that is provided.

3. The primary usage of Bayesian methodology is to provide uncertainty quantification for small data sizes. In that respect, I feel it is unfair to compare the current method to Bayesian approaches.

4. How do the approximation of centroids vary ? For example if you have a mixture of a Cauchy and a Gaussian distribution, the modes of the distributions converge at different rates. As a result some more noisy data may not even reveal an interpretable centroid unless the datasize is sufficiently large. How do you plan to alleviate this issue?



**Summary Of The Paper:**

The paper proposes a way to optimise the choice of Bootstrap samples so as to obtain efficient inference with smaller number of such samples. While traditional Bootstrap requires the need to use a large number of samples to approximate the target distribution, the authors propose a combination of k-means type algorithm is association with Bootstrapping to achieve the goal. In addition, they provide a theoretical justification for the method along with providing some experimental evaluation .

**Summary Of The Review:**

I think the paper is understandable and has a nice idea though not novel and needs a lot of clarification and revision before being accepted for publication

---

> ### Author Response · Authors · 2021-11-22
> **Thanks reviewer 5nLX for the comments. Below please find our response.**
>
> **Regarding the novelty**
> We think the key contribution of this paper is not using k-means to improve the sample quality but on *how to do k-means when you are actually not able to access the target distribution* (from a very high-level abstract perspective). This is also the key differentiator of this paper to the corsets paper you suggest. The corsets-based approach requires us to have the log pdf of the target distribution or assume it's easy to sample from the target distribution.  These two requirements can not be satisfied in our case. In the updated manuscript, we add a new paragraph right above section 3.1 to explain why our approach is different from those coresets based approaches.
>
> Also, as the key contribution is not k-means but ''k-means without accessing target distribution'', we feel the classical testing on mixture model you suggest is not that related to our approach.
>
> **Regarding theorem 2**
> Theorem 2 holds for any $\pi$ that satisfies our assumption 2. Intuitively, we need a $\pi$, whose resampling of the empirical distribution still well approximates the true population distribution when $n$ is very large.
>
> **Regarding Eq.5**
> As mentioned in the paper (footnote 3), the formal characterization on why Eq.5 holds is in Theorem 1, in which it shows that Eq.5 holds with high probability for $w$ (not almost surely for w).
>
> **Regarding the intuition of the theorem**
> When n is very large, the empirical distribution can well represent the true distribution. And hence the resampling distribution (i.e. sample n i.i.d. data point with replacement from the empirical distribution) also well represents the empirical distribution. This gives that the bootstrap resampled distribution well represents the true distribution and hence we have $L_w \approx L_\infty$. As the losses are close, their minimizers are also close (hence $\hat{\theta}_w \approx \theta_0$). We also update the manuscript to give more illustrations of the intuitions (see the newly add sentences below equ (5)).
>
> **Regarding comparing with Bayesian approach**
> We are not sure how to interpret this comment. We aim to improve the scalability of the frequentist uncertainty quantification approach. In the manuscript, we didn’t compare with the Bayesian approach.
>
> **Regarding the approximation of centroid**
> Thanks for raising up these comments. In this paper, we assume the tail behavior of the target distribution is good (i.e. asymptotic normality). Again, as mentioned, the key contribution is not how to improve clustering but "use the k-means type of idea to optimize the particles when we are not able to really access the target distribution". We agree that the question you have is an interesting research problem but we do feel it is far out of the scope of this submission.

---

### Official Review · Reviewer_y4AP · 2021-11-03

**Correctness:** 2
**Technical Novelty And Significance:** 3
**Empirical Novelty And Significance:** 2
**Recommendation:** 3
**Confidence:** 4

**Details Of Ethics Concerns:**

Please see pg 8, Section 5.3: "Efficient exploration is a major challenge for reinforcement learning (RL). Common dithering strate- gies such as ε-greedy do not carry out temporally-extended exploration, which leads to exponentially larger data requirements (Osband et al., 2016)."

This excerpt from the paper under review is plagiarized from the first two sentences of the abstract of Osband et al, 2016. I don't suspect nefarious intent since the authors cite Osband et al, 2016 directly after plagiarizing the text. However, the authors must rephrase or quote (rather than copy-paste) and cite any material drawn from another source.

===============

Update 11/29: This issue was addressed by the authors in the discussion period, so I no longer have any concerns as such.

**Main Review:**

**Strengths:**

- This work seeks to improve on computational aspects of the bootstrap, a fundamental and popular statistical method

- The derivation of the centroid method involves interesting ideas from optimal transport and a natural choice of surrogate loss.

- The applications seem well-chosen and relevant to modern machine learning

**Concerns:**

- I could not understand how the centroid method is applied in the contextual bandit, bootstrap DQN, and ensemble learning experiments. In the "Contextual bandits" section below I have several questions and comments that I hope will clear up my confusion and suggest ways to improve the presentation.

- The training overhead should not be completely ignored since the main motivation of this paper is computational. It would strengthen the paper to make note of the training overhead for all experiments in addition to the note in Appendix C.6.

- Although the topic of this paper is bootstrapping, there is no evaluation of uncertainty for the real-data experiments.

- The presentation of the algorithm in **Practical algorithm** is difficult to follow because the run-time analysis and algorithm description are all written together. One suggestion to improve readability is to move the Algorithm Box here, give a short description of the practical algorithm in words, and then justify its running time.

- The derivation and theory are essentially limited to concave log-likelihoods (otherwise, the Wasserstein distance here is not a proper distance), though the algorithm may be implemented for general losses

- Bootstrap DQN is limited to 2 and 5 models. This makes it hard to see trends.

**Contextual bandits**

- It seems that each model has a parameter (that of a neural net), and so the number of models is exactly the number of particles. Is that correct? This was not clear to me from the text.

- It seems that the centroid algorithm applied to the collection of parameters of the models. Is this correct? This should be explicitly stated.

- If the previous bullet is correct, what datapoints are input to the centroid algorithm. Is it only the datapoints in the memory buffer? Or a resampling according to $w$? What datapoints are used for the centroid degeneration-prevention step?

- Is the centroid method applied after every 50 contexts? Further, does one first learn the parameters of each model using SGD and then use these learned parameters as the initialization for the centroid method?

- Once answered, these points can be included in the main text. More generally, for each experiment, a precise step-by-step explanation of how the centroid method is used would greatly improve readability.

**Further comments**

- The paper *Wasserstein measure coresets* of Claici, Genevay, and Solomon considers a similar methodology with a different loss function and different applications from this paper. In general, the constructions in the paper under review are reminiscent of the literature on coresets (cf *Introduction to Core-sets: an Updated Survey* by Feldman).

- pg 3: Equation (3) is abrupt and not explained. It seems to rely on some important theorems from optimal transport (cf Lemmas 3.1 and 3.2 in Canas--Rosasco '12). More justification here is needed.

**Minor issues**

- In the Tables 1,2,3, please write the value of the best performing method in boldface for readability.

- pg 5, typo: estiblish

- pg 5, "... can be done using classic techniques in non-convex optimization..." This needs further elaboration or should be removed.

- pg 6: Theorems 1 and 2 are asymptotic in nature, and this should be mentioned again in the theorem statements

- pg 7, "Asymptotics when $m$ grows": The discussion below Proposition 7 of Weed--Bach '19 implies that for absolutely continuous measures, the iid particle measure approximately attains the rate $m^{-1/d}$, which is minimax optimal. Thus as $m \to \infty$, it is not possible to beat iid sampling. This is also supported by Figure 2.

- pg 8, Figure 3: Interestingly, there is a period where centroid 2-head outperforms centroid 5-head. Why does this happen?

- pg 9, "However, those approaches are not applicable..." This is not true. The methods of Chen et al (and the coreset literature more broadly) work for a given dataset.

- pg 10: Please used published versions in the bibliography when possible, not just arxiv links. Also the phrase "et al" appears in the bibliography and should be removed.

- pg 13, Proof of Theorem 1: Please give a reference for the claim that the MLE is subGaussian. This should require some assumptions (note: it is not implied by asymptotic normality).

- pg 16, Algorithm 1: Why is there a loop over t' in inner-iters? I did not see any dependence on t'.

- pg 16, Methods to construct confidence interval: the $\hat \rho_\pi$'s should be $\tilde \rho_\pi$'s



**Summary Of The Paper:**

This paper aims to improve the quality of the parametric bootstrap when the number of particles is limited. It does so by constructing a small collection of "centroids" that well-approximate the ideal bootstrap distribution on parameter space in a certain Wasserstein distance. Practically, the centroids are constructed by gradient descent with respect to a loss function involving the negative log-likelihood. Theoretically, this paper shows that the surrogate loss converges to the Wasserstein distance under some assumptions. Empirically, this paper applies the centroid methodology to confidence interval construction, contextual bandits, the bootstrap deep q-network, and ensemble learning .

**Summary Of The Review:**

I like the topic and motivation of this paper, and the derivation of the method is interesting. However, the most important part of this paper is the evaluation. This paper needs to explain better how the algorithm is applied in real-data scenarios. Until it can be clarified what precisely is happening in the experimental sections, it is difficult for me to judge effectiveness of this method. I am also concerned that there are no uncertainty estimates in the real-data experiments and only a few run-times are reported. It seems that careful revision is needed for this paper to be ready for publication, and that is the main reason for my score.

========================================

**Update on review (11/29)**

The algorithm used in the real-data experiments seems to differ in a significant way from the centroid method derived in Section 3 (see my discussion with the authors). Further, these differences can only be ascertained by a careful read of the Appendix, which leads to a lack of clarity on what is being proposed. I think this paper needs a better explanation of how the methodology used in the real-data experiments matches with the initial derivation of the centroid method. I do think that both algorithms are quite interesting and that proper attention to both of them would improve this paper.

I also think that more care should be taken to report the training times for all of the experiments. I understand that training time is not the main focus of the paper, but carefully reporting run-times/overheads would clarify to what extent the centroid method is practical.

Overall, I think that improving the quality of bootstrap is a promising idea that should be further explored, and this paper takes an interesting route toward doing so. However, I feel that this work needs to be further developed to be ready for publication, so I have kept my initial scores the same.

---

> ### Author Response · Authors · 2021-11-22
> **Thanks reviewer y4AP for the very detailed comments. Below please find our response.**
>
> **Regard questions related to contextual bandits**
> Thanks for the comments! We agree that our presentation was a bit unclear. We’ve updated the manuscript accordingly, in which detailed information is given to address all your concerns. We think directly looking at the updated context (C2.2 in appendix) might better answer your question.
>
> 1. Yes, in the context of this paper, bootstrap sample/particle/centroid means a machine learning model. We add a sentence in the introduction to explicitly state this for clarification.
>
> 2. Yes, as each model is a particle, the centroid algorithm is applied to the collection of parameters.
>
> 3. For each centroid, if it does not degenerate, its’ training set (replay buffer) is a resampled set of all the observed contexts. The resampling probability is calculated by $\boldsymbol{w}_h$. If it degenerates, its’ training set (replay buffer) is just all the observed context.
>
> 4. For the exact procedure, we refer you to the C2.2 in the updated manuscript.
> The replay buffer, parameter, and $v_j^*$ for each model are updated every 50 contexts. In the contextual bandit experiment, we initialize the centroid by drawing from a gaussian distribution.
>
> **Regarding the training cost**
> As mentioned in footnote 2, we mainly aim to improve performance given the limited inference budget, and hence the main focus is not reducing the training time (since it’s usually negligible compared with the inference cost).
>
> **Regarding evaluate the uncertainty in real data**
> Our experiments (except for the confidence interval) are all based on standard benchmarks. For the confidence interval experiment, it seems only can be using some simulated data as only in this way we are able to evaluate the performance by calculating the coverage probability.
>
> **Regarding the practical implementation**
> Thanks for the comments. We’ve updated the manuscript accordingly. Due to the space constraints, it’s hard to move the algorithm boxes to the main text. But in the appendix, we now provide both ideal implementation and practical implementation with more detailed information listed. We hope this can improve the clarity.
>
> **Regarding the concave log-likelihood**
> Yes, we admit that currently, we can only provide the justification when assuming the dataset is large and hence asymptotic normality holds. But we think this is the most common case in machine learning applications
>
> **Regarding m in Bootstrap DQN**
> m=2,5 is actually a very common design in DQN (in practice, we seldom use more than 5 heads). Since we focus on the improvement when m is small, we think such design is proper. And in both cases, centroid approach is better than bootstrap.
>
> **Regarding Wasserstein measure coresets**
> Thanks for the literature. From a high level abstracted perspective, our paper is on ‘how to use k-means type of approach to improve the particle quality without really accessing to the target distribution’. This is a key differentiator of our paper to other papers that improve the sample quality, as they all need to access to the target distribution (know the log pdf or assume we can efficiently sample from target distribution.) We add a paragraph (right above section 3.1) to discuss the difference between our method and other method that improves the sample qualities.
>
> **Regarding equ (3)**
> Thanks for the comments. Yes, it’s a direct application of Lemma 3.1 and 3.2 in [1] and we update the manuscript adding a reference.
>
> **Regarding the ethics issue**
> We apologize for the issue and appreciate that you point this out. Yes, we don’t mean to plagiarize the sentence but didn’t realize that a quote is needed in this case. Correction has been made.
>
> **Regard the minor issues**
> Thanks for all those comments. Typo/format issues are all fixed in the updated manuscript.
>
> We update theorem 1,2, with a sentence explicitly states the asymptotic nature of our theorem.
>
> The i.i.d. sample gives the minimax bound for the convergence of expected Wasserstein distance. For the convergence rate in probability, the rate is different. See Theorem 5.1 in [1].
>
> Regarding Figure 3: We think the reason is that there is a region that the model is basically improving by exploiting. The m=2 gives less uncertainty and hence the model benefit from exploiting in such a period.
>
> Please see our response on corset based approach in “Regarding Wasserstein measure coresets”
>
> Regarding subGaussian MLE: Thanks very much for pointing this out! Yes, MLE is not subGaussian and we made a correction according. We should change the statement to: for any $r,\epsilon>0$, when $n$ is sufficiently large, $P(||\hat{\theta}_w - \theta_0||>r)\le \exp(-\lambda n r) + \epsilon$. Note that the result of the main theorem remains the same as we only claim high probability bound.
>
> Regarding the algorithm 1: We updated the algorithm box and hope this improve the clarity.
>
> [1] Canas, Guillermo D., and Lorenzo Rosasco. "Learning probability measures with respect to optimal transport metrics.

---

> > ### Comment · Reviewer_y4AP · 2021-11-24
> > **Re:**
> >
> > I appreciate the detailed response and revision. I will take them into account when updating my review. I think that making the algorithm boxes for Algorithms 3 and 4 was a nice choice and improves the presentation.
> >
> > I would like to ask for another clarification on Algorithm 3. It seems that lines 9--17 do the following for a given model/particle j: (i) use the current parameter for model j to make a *new distribution* over the contexts, (ii) sample from this distribution to get a new context set $R_j$ for model j, (iii) train the model using RMSprop on $R_j$ to get new parameters. Here the *new distribution* over contexts in (i) is such that a context is weighted by a stochastic version (15) of the gradient step (10) from the centroid algorithm, Algorithm 2.
> >
> > Is this description accurate? If so, I wonder if you could explain how Algorithm 3 relates to the original centroid algorithm. In particular, what is the intuition for reweighting contexts by the gradient steps of the centroid algorithm, Algorithm 2? The way the $\theta_j$'s evolve in each algorithm seems to be quite different, and I'm not sure how the method derived in Section 3 can be used to capture Algorithm 3. I do see how the weights from Algorithm 2 are used to weight the models/particles in Algorithm 3 though.
> >
> > Also a more minor point, I think that $w_{h,i}$ was not defined in Section B. Could you clarify what this is?

---

> > > ### Author Response · Authors · 2021-11-24
> > > **Thanks reviewer y4AP for the follow-up question**
> > >
> > > Thanks reviewer y4AP for the follow-up question!
> > >
> > > Sorry we have a typo on the definition of $q_{i,j}$. The correct one should be
> > > $q_{i,j}=\frac{\sum_{h=1}^{M}I(j\in u_{w_{h}})w_{h,i}}{\sum_{h=1}^{M}[I(j\in u_{w_{h}}=j)]}$ (no summation over i on the numerator). It is the Monte Carlo approximation of $E_{w}(w_{i}\mid u_{w}=j)$ (not the gradient step (10)). $q_{i,j}$ can be viewed as, "among all those w that is assigned to centroid j, the average of the resampling weight of the i-th data point".
> > >
> > > Thus the algorithm 3 is consistent to Algorithm 2. The only difference is that: algorithm 2 is applied to an offline case and the reweighting of context is on the whole dataset.  Algorithm 3 is for an online case and the reweighting of context is on the whole observed dataset at the current iteration.
> > >
> > > $ w_{h,i}$ is the i-th element of $w_h$. Thanks for pointing this out. We will correct the typo of $q_{i,j}$ and explicitly define the indexing of vector in the next version.

---

> > > > ### Comment · Reviewer_y4AP · 2021-11-24
> > > > **Re: follow-up question**
> > > >
> > > > Thanks for the response! The interpretation of $q_{ij}$ was helpful. I have one more question for now. Can you explain how $\theta_{j}^*(t)$ evolves in Algorithm 3? Is it via $\theta_j^*(t+1) \leftarrow \theta_j^*(t) - \varepsilon \phi(\theta_j^*(t))$ where $\phi$ is defined in (13), or is it via RMSprop, or some combination of the two?

---

> > > > > ### Author Response · Authors · 2021-11-25
> > > > > **Response**
> > > > >
> > > > > Thanks for the point. The final updating is through the RMSprop. But the gradient is calculated using (13). More specifically, after line 9-18, each centroid now has its own updated buffer (training set). If a centroid degenerates, then its buffer would simply be all the data points that we observed, otherwise it would be the reweighted version.
> > > > >
> > > > > During the training, for each centroid, we sample a mini-batch of data from its own buffer, calculate the loss and call `loss.backward()` and then `optimizer.step()`, where optimizer is the RMSprop.
> > > > >
> > > > > Thanks!

---

> ### Author Response · Authors · 2021-12-06
> **Thanks for the update**
>
> Thanks Reviewer y4ap for all the discussion and comments.
>
> We agree that this submission needs further improvement.
>
> If possible, could you give more detailed feedback on what's your current main concern, besides the lack of a more detailed training time comparison? We would appreciate your feedback and will improve the manuscript based on that.
>
> Thanks again!

---

> > ### Comment · Area_Chair_UUmv · 2021-12-06
> > **response to authors**
> >
> > I will be incorporating feedback also given in private comments during reviewer discussion in the meta-review.  Thanks for your active participation in the review and discussion process, and willingness to incorporate reviewer feedback.

---

### Author Response · Authors · 2021-11-22
**Summary of Revision**

We thank all the reviewers for their constructive comments!

We've updated the manuscript based on the reviewers' comments and we summarize the main updates here. All the changes are written in red to make the tracking easier.

1. We add a sentence to clarify that we might directly call a machine learning model particle/centroid/sample given the context of this submission.

2. We add a highlight at the beginning of this paper, emphasizing we aim to improve the bootstrap when the resource at *inference* rather than *training* is limited.

3. We add references for why (3,4) returns the global minimizer.

4. We give a more detailed illustration on why equ (5) holds.

5. We add a new paragraph to discuss the difference between this paper and other corset-based approach that also improves the particle quality.

6. We reorganize the 'Practical algorithm' subsection to improve the readbility.

7. We clarify that the setting of our theory is on the fixed parameter dimension.

8. We add a sentence in the statement of the main theorem to emphasize that our theorem is asymptotic.

9. We give extensive revision to the algorithm box part (Appendix B). Showing the ideal algorithm and practical implementation with improved readability. Also, we give the detailed procedure when applied to contextual bandit and reinforcement.

10. Experiment details are further refined.

11. Typo/Format issues are fixed.

---

### Decision · Program_Chairs · 2022-01-20

**Decision:**

Reject

**Comment:**

The submission aims to improve the quality of the bootstrap when the number of samples is small.  It does so by gradient descent on the to approximate the ideal bootstrap in Wasserstein distance.  The submission combines a nice set of methodologies, and aims to address an interesting statistical problem in a principled way.  The reviewers were unanimous in their opinion that the submission falls below the threshold for acceptance to ICLR.  It was revealed in post rebuttal discussion with reviewer y4AP that they wish to retain a reject recommendation due to a lack of clarity in the methodology even after author comments.  The review details specific issues that can eventually be clarified in a revision for submission to another venue.